# Dissection of *Besnoitia besnoiti* intermediate host life cycle stages: From morphology to gene expression

Chandra Ramakrishnan[1]*, Aarti Krishnan[2,¤a,¤b], Samuel Francisco[3], Marc W. Schmid[4], Giancarlo Russo[5,¤c], Alexandre Leitão[3,6], Andrew Hemphill[7], Dominique Soldati-Favre[2], Adrian B. Hehl[1]*

**1** Parasite Cell Biology Laboratory, Institute of Parasitology, Vetsuisse Faculty, University of Zurich, Zurich, Switzerland, **2** Department of Microbiology and Molecular Medicine, University of Geneva, CMU, Geneva, Switzerland, **3** CIISA—Centro de Investigação Interdisciplinar em Sanidade Animal, Faculdade de Medicina Veterinária, Universidade de Lisboa, Lisbon, Portugal, **4** MWSchmid GmbH, Glarus, Switzerland, **5** Functional Genomics Center Zurich, Zurich, Switzerland, **6** Laboratório Associado para Ciência Animal e Veterinária (AL4AnimalS), Lisbon, Portugal, **7** Institute of Parasitology, Vetsuisse Faculty, University of Bern, Bern, Switzerland

¤a  Current address: Institute for Medical Engineering & Science and Department of Biological Engineering, Massachusetts Institute of Technology, Cambridge, Massachusetts, United States of America
¤b  Current address: Infectious Disease and Microbiome Program, Broad Institute of MIT and Harvard, Cambridge, Massachusetts, United States of America
¤c  Current address: EMBL Partner Institute for Genome Editing, Life Sciences Center–Vilnius University, Vilnius, Lithuania
* chandra.ramakrishnan@uzh.ch (CR); adrian.hehl@uzh.ch (ABH)

**Data Availability Statement:** All DNA sequencing files are available at NCBI's Sequence Read Archive (SRA) under Bioproject PRJNA386239 (https://

## Abstract

Cyst-forming Apicomplexa (CFA) of the Sarcocystidae have a ubiquitous presence as pathogens of humans and farm animals transmitted through the food chain between hosts with few notable exceptions. The defining hallmark of this family of obligate intracellular protists consists of their ability to remain for very long periods as infectious tissue cysts in chronically infected intermediate hosts. Nevertheless, each closely related species has evolved unique strategies to maintain distinct reservoirs on global scales and ensuring efficient transmission to definitive hosts as well as between intermediate hosts. Here, we present an in-depth comparative mRNA expression analysis of the tachyzoite and bradyzoite stages of *Besnoitia besnoiti* strain Lisbon14 isolated from an infected farm animal based on its annotated genome sequence. The *B. besnoiti* genome is highly syntenic with that of other CFA and also retains the capacity to encode a large majority of known and inferred factors essential for completing a sexual cycle in a yet unknown definitive host. This work introduces *Besnoitia besnoiti* as a new model for comparative biology of coccidian tissue cysts which can be readily obtained in high purity. This model provides a framework for addressing fundamental questions about the evolution of tissue cysts and the biology of this pharmacologically intractable infectious parasite stage.

www.ncbi.nlm.nih.gov/sra/?term=PRJNA386239).
All RNA-Seq files (fastq.gz) are available from
NCBI's SRA with accession numbers SRS2223086
for bradyzoites (https://www.ncbi.nlm.nih.gov/sra/
?term=SRS2223086) and SRS2223087 for
tachyzoites (https://www.ncbi.nlm.nih.gov/sra/?
term=SRS2223087).

**Funding:** This work was supported by the Swiss
National Science Foundation, SNSF Sinergia grant
(CRSII3_160702 to ABH and DS) and the SNSF
grant (310030_184662 to AH) https://snf.ch/en/
FKhU9kAtfXx7w9AI/page/home. The funders had
no role in study design, data collection and
analysis, decision to publish, or preparation of the
manuscript.

**Competing interests:** The authors have declared
that no competing interests exist.

## Author summary

The unicellular parasite *Besnoitia besnoiti* causes besnoitiosis in cattle. This emerging disease is characterized by a chronic phase where the parasite resides in tissue cysts mainly in the skin. Besnoitiosis leads to significant economic losses due to abortions, reduced milk production and/or leather quality and infertility in bulls. Transmission is known to take place by direct contact of infected animals or parasite transfer by blood-sucking insects. However, a definitive host in which sexual development takes place has not yet been identified. Here, we provide a detailed microscopical characterization of the parasite, compare the genome of *B. besnoiti* to related parasites, reveal the gene expression profiles of acute and chronic stage parasites, and show that the *B. besnoiti* genome contains genes necessary for sexual development in an as yet unknown definitive host. Our study provides a deeper understanding of *B. besnoiti* biology, highlights unique features of chronic cyst stages, and discusses its potential as a model organism for other related tissue-cyst forming parasites.

## Introduction

Bovine besnoitiosis caused by *Besnoitia besnoiti* is an (re-)emerging disease in Southern Europe (reviewed by Álvarez-García *et al.* [1]) resulting in considerable economic losses in the cattle industry. Bovine besnoitiosis is responsible for decreased milk production, abortions, bull infertility and reduction of hide quality (reviewed by Álvarez-García *et al.* [1] and Cortes *et al.* [2]). Thus, besnoitiosis has gained interest in the search for effective therapeutics. Despite this interest, the biology of this parasite remains largely unknown. *B. besnoiti*–a tissue cyst-forming Apicomplexa (CFA)—is genetically related to *Neospora caninum* and *Toxoplasma gondii* but with clear differences in transmission and uncertain definitive host cycle. Transmission is known to take place via close contact (e.g. mating) or horizontally via mechanical transmission by blood-sucking insects (reviewed by Álvarez-García *et al.* [1] and Cortes *et al.* [2]). Vertical transmission from mother to the foetus–an important route for *T. gondii* and *N. caninum*—has never been observed for *B. besnoiti* [3]. Thus, as for neosporosis and in contrast to toxoplasmosis, herd management is an important factor in controlling the spread of the besnoitiosis. However, to date, no definitive host in which sexual development can take place has been identified. Studies on wild, domestic and laboratory animals have been conducted to discover the definitive host [4–7], but no conclusive identification could be made.

In contrast to other described CFA, which build walled cysts in muscle and brain, *B. besnoiti* forms large tissue cysts in cells of the mesenchymal lineage and are prevalent in the dermis, sclera and mucosa. This peripheral niche is conducive to transmission by insects or direct mechanical transfer from chronically infected cattle to a new host. With a diameter of up to 0.5 mm, these cysts are macroscopically detectable. The tissue cysts are enclosed by three distinct layers comprised of host-derived connective tissue surrounding the infected host cell, and a parasite-derived cyst wall on the luminal face of the parasitophorous vacuole membrane (PVM) containing the bradyzoites [8]. Tissues with high cyst loads, in particular the dermis, show altered architecture, sometimes with significant pericystic leukocyte infiltration and compromised barrier integrity [9, 10]. The superficial, and to some extent also the deep dermis, appear expanded by infiltration of immune cells, e.g. macrophages and eosinophilic granulocytes, and shows distinct signs of fibrosis [11, 12]. Tissue cyst formation affects animal health and agricultural productivity. Acute phase tachyzoite stages can be cultured *in vitro* in fibroblast host cells but unlike for *T. gondii*, there is currently no protocol allowing induction

of differentiation to bradyzoites and cyst formation, which precludes systematic molecular genetic characterization of this infectious life cycle stage.

In this study, we performed whole genome sequencing and annotation of *B. besnoiti* and identified genes that are predicted to be necessary for the development of gametes and oocysts in a putative definitive host. Deep comparative RNA-Seq analysis of bradyzoites extracted from *in vivo* tissue cysts and tachyzoites cultured *in vitro* reveal strikingly distinct stage-specific transcriptomes with many similarities to those of the corresponding *T. gondii* stages.

## Results and discussion

### Isolation and characterization of tissue cysts

One 11 years old Charolais cow and one 7 years old Limousin cow both with chronic besnoitiosis were used in this work. Both animals were severely affected and presented a very high density of cysts in the dermis. Subcutaneous tissue blocks from the dermis were transferred to a Petri dish and tissue cysts–readily visible with a binocular (Fig 1A) or by conventional light microscopy (Fig 1B)–were released from the surrounding tissue by scraping the blocks with a needle and rinsing with saline solution. Semi-thin sections of epoxy resin-embedded cysts stained with methylene blue and basic fuchsin (Fig 1C) revealed a densely packed and dark blue interior compartment, and a tissue cyst wall of approximately 40–45 μm thickness. Scanning transmission electron microscopy (SEM, Fig 1D–1F) showed that cysts were either covered with connective tissue, or they exhibited a seemingly smooth surface structure. However, a higher magnification view of the cyst surface revealed the presence of densely packed layer of filaments or filament bundles covering the entire surface (Fig 1G). Passing these tissue cysts through an 18-gauge syringe needle resulted in cyst rupture and the release of bradyzoites with associated matrix material into the surrounding medium (Fig 1H and 1I).

Transmission electron microscopy (TEM) analysis showed that the tissue cysts consisted of several distinct layers of host and parasite origin, namely an outermost cyst wall enriched in collagen fibers, surrounding the host cell plasma membrane and cytoplasm, an inner cyst wall from parasite material deposited on the luminal side of the parasitophorous vacuole membrane, and the cyst interior lumen with bradyzoites embedded in the cyst matrix (Fig 2A). The outer tissue cyst wall was composed of host-derived extracellular fibrous material, most notably collagen-like filaments that were partially assembled into bundles, more densely packed at the exterior parts and more loosely closer to the host cell surface (Fig 2A–2C). A thin (2–5 μm) layer of host cell cytoplasm surrounded the entire tissue cyst and contained many electron-dense mitochondria. In addition, the host cell cytoplasm was largely filled with small filaments and filament bundles, and numerous cytoplasmic extensions into the outer cyst wall were visible (Fig 2D–2F). The intracytoplasmic or inner cyst wall had a thickness of 0.4–0.7 μm and appeared as an electron-dense entity composed of a granular ground substance, aggregated on the luminal side of the PVM (Fig 2G). Vesiculated structures, mainly filled with electron dense material, were often observed in the vicinity of the inner surface of the cyst wall; in contrast, such vesicles were not found in the cyst interior (Figs 2G, 3B and 3E).

A large number (> 500) bradyzoites were embedded into the matrix of the inner lumen of the tissue cyst. With few exceptions (marked by an arrow in Fig 3A), most parasites appeared viable, and TEM showed the hallmark features of Apicomplexa such as an apical conoid [13, 14], and numerous secretory organelles such as micronemes, dense granules and rhoptries [15] (Fig 3B–3E). Features that were consistent with the subcellular morphology of bradyzoite-stage apicomplexans included the positioning of the nucleus in the posterior region of the cell (Fig 3C), and a high number of amylopectin granules [16, 17] located within the parasite cytoplasm (Fig 3A–3E). In the more peripheral regions of the tissue cyst, bradyzoites were tightly

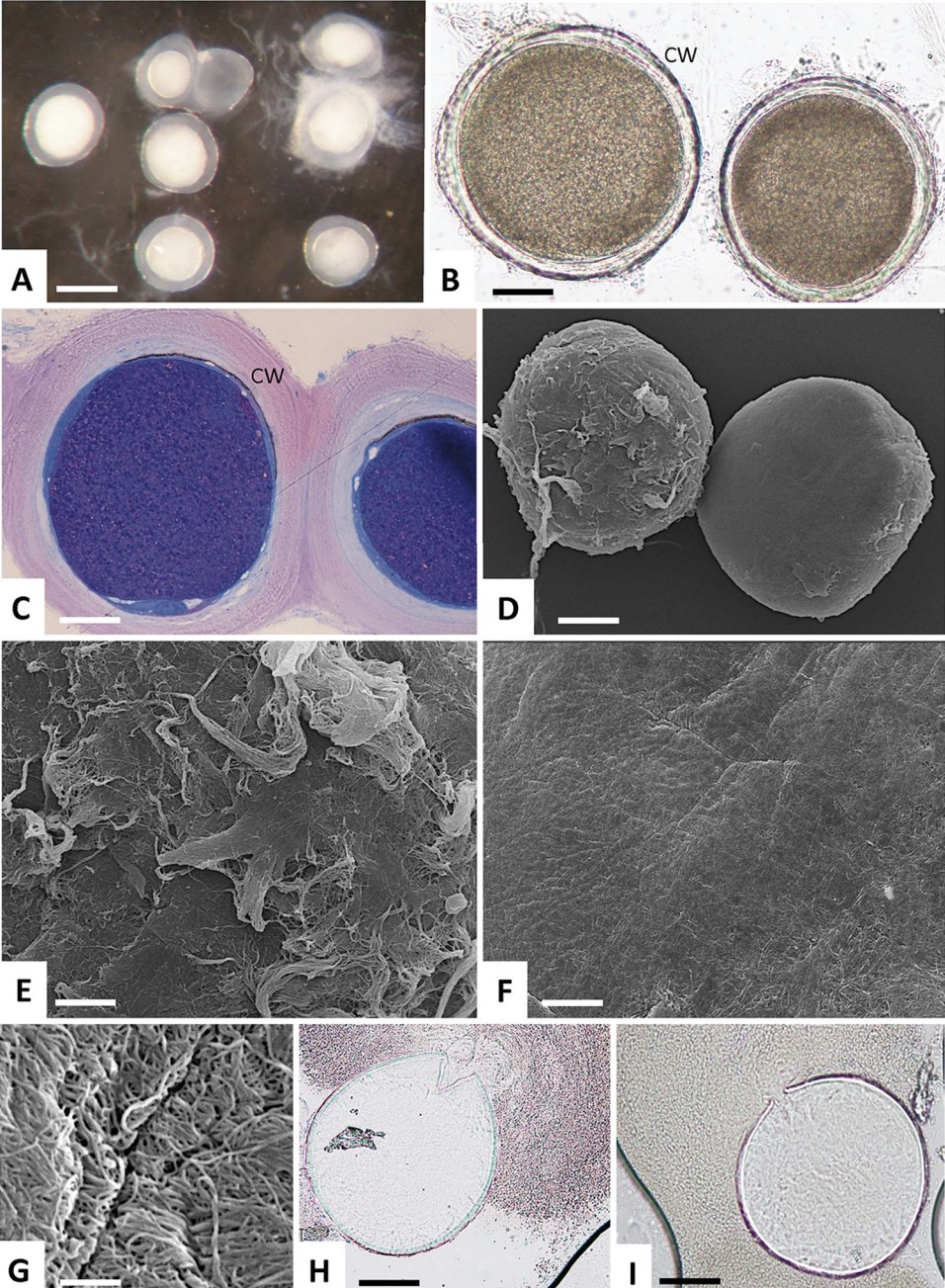

**Fig 1. Microscopical description of isolated *B. besnoiti* tissue cysts.** (A) Binocular view of cysts freshly isolated from skin. (B) Conventional light microscopy. CW = cyst wall. (C) Light microscopy of semi-thin sections of epoxy-resin embedded cysts stained with methylene blue and basic fuchsin. (D-G) Scanning transmission electron microscopy of the tissue cyst wall. (H, I) Ruptured cyst releasing bradyzoites and cyst matrix. Bars in A = 500 μm, B = 140 μm, C = 110 μm, D = 170 μm, E and F = 40 μm, G = 1.6 μm, H and I = 140 μm.

packed (Fig 3A–3E) and embedded in the intravacuolar network (IVN) composed of meshwork of membranous filaments, best seen in Figs 1G, 3A and 3B. In contrast, bradyzoites in the interior and more central areas of the cysts were less numerous and much further apart from each other and embedded in a granular matrix filling the intercellular space (Fig 3F). Viability of bradyzoites and their distribution within tissue cysts have not been assessed.

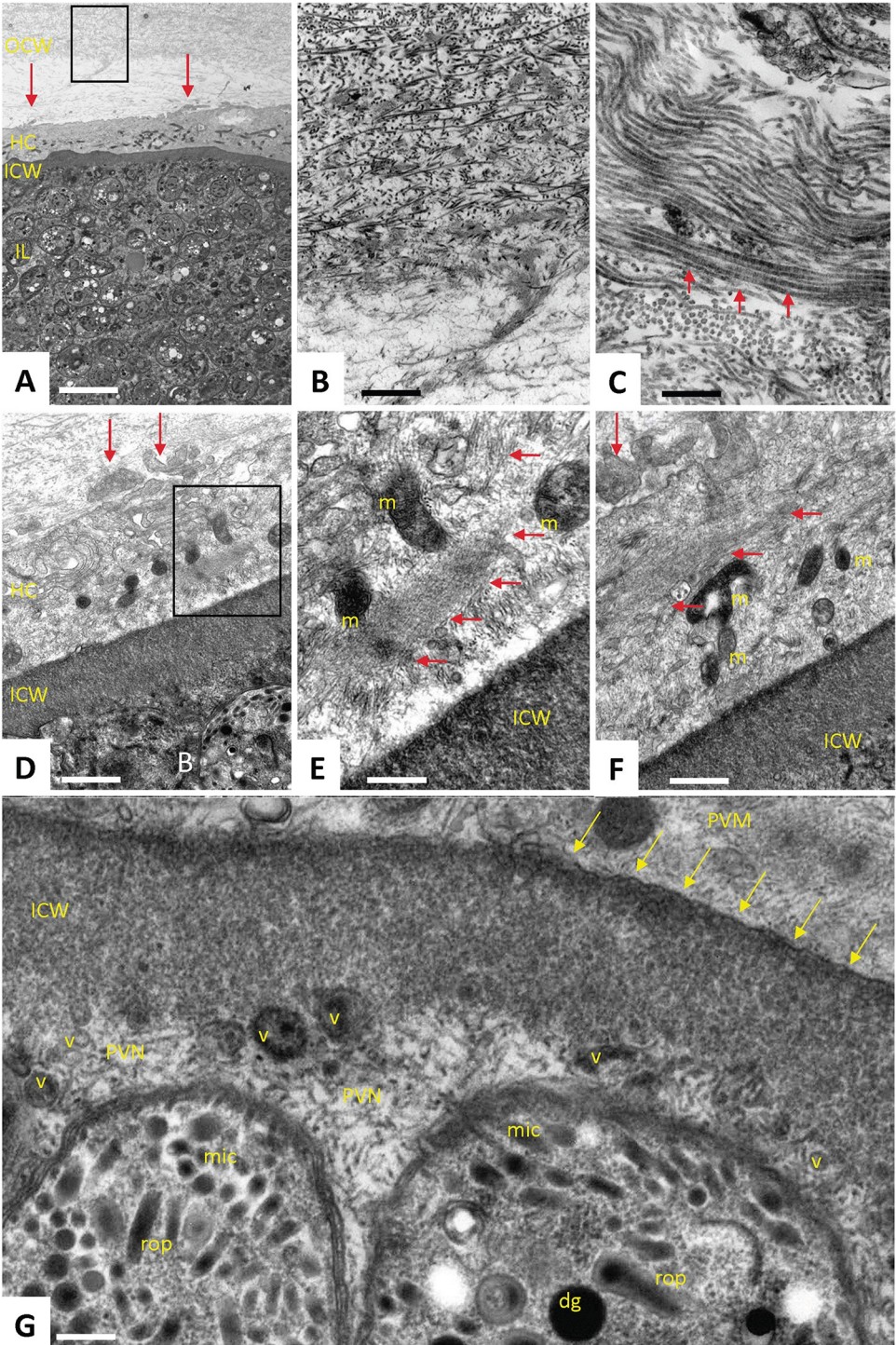

**Fig 2. Transmission electron microscopy of tissue cysts. (A-C)** Compartments of the tissue cyst, with the fibrous outer cyst wall (OCW), host cell cytoplasm (HC), inner cyst wall (ICW) and the inner lumen of the tissue cyst (IL). Red arrows indicate peripheral cytoplasmic host cell-extensions into the OCW. The boxed area in **(A)** is magnified in **(B)**, showing that the OCW is composed largely of fibrous material, and in **(C)** an even higher magnification depicts the collagen-like filament bundles (red arrows). **(D)** Host cell cytoplasm with cytoplasmic extensions (vertical red arrows) and inner cyst wall. The boxed area in (D) is enlarged in (E). **(E-F)** Larger magnification of the host cell cytoplasm adjacent to the inner cyst wall, with the presence of numerous electron-dense mitochondria (m) and many cytoskeletal filament bundles (horizontal red arrows). **(G)** Higher magnification view of the parasite-host cell interface, showing the ICW and its electron-dense granular substance, the parasitophorous vacuole membrane (pvm, yellow arrows),

distinct vesicular structures adjacent to the ICW interior. Two bradyzoites are embedded in a parasitophorous vacuole network (PVN), exhibiting their apical parts with micronemes (mic), rhoptries (rop) and dense granules (dg).Bars in A = 2.8 μm, B = 0.55 μm, C = 0.25 μm, D = 0.55 μm, E = 0.25 μm, F = 0.35 μm, G = 0.15 μm.

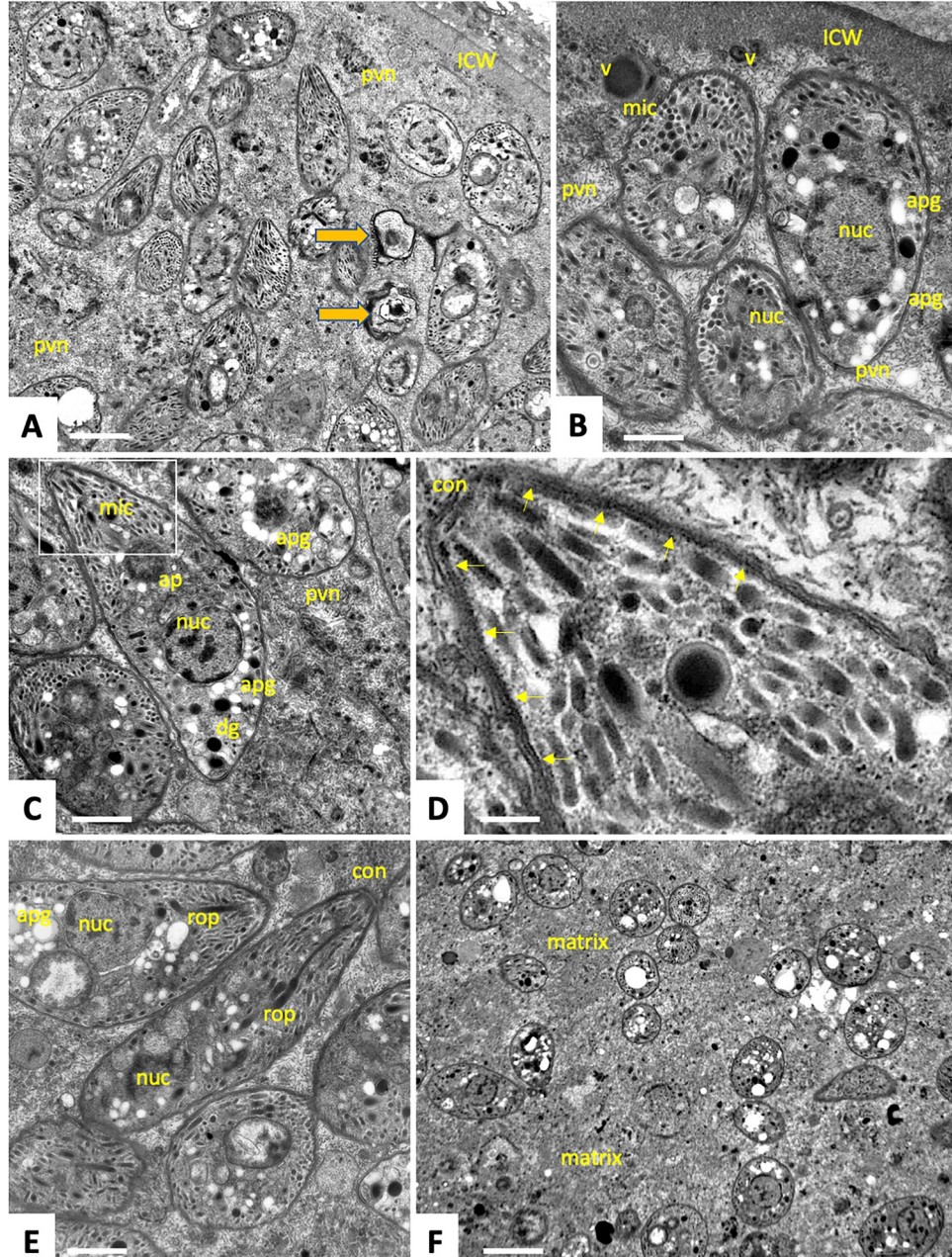

**Fig 3. Transmission electron microscopy of the cyst interior and bradyzoites. (A)** Bradyzoites embedded in the tissue cyst matrix in the peripheral area of the cyst. Non-viable bradyzoites are marked with yellow arrows. **(B-E)** Higher magnification view of bradyzoites with typical zoite organelles. (D) enlarged inset from (C) showing details of the conoid (con) and the inner membrane complex, IMC (arrows). **(F)** Bradyzoites in the centrally located area of the tissue cyst. apg: amylopectin granules, con: conoid, dg: dense granules, ICW: inner cyst wall, mic: micronemes, nuc: nuclei, pvn: parasitophorous vacuole network, rop: rhoptries, v: vesiculated structures. Bars in A = 1.2 μm, B = 0.5 μm, C and E = 0.6 μm, D = 0.15 μm, F = 1.4 μm.

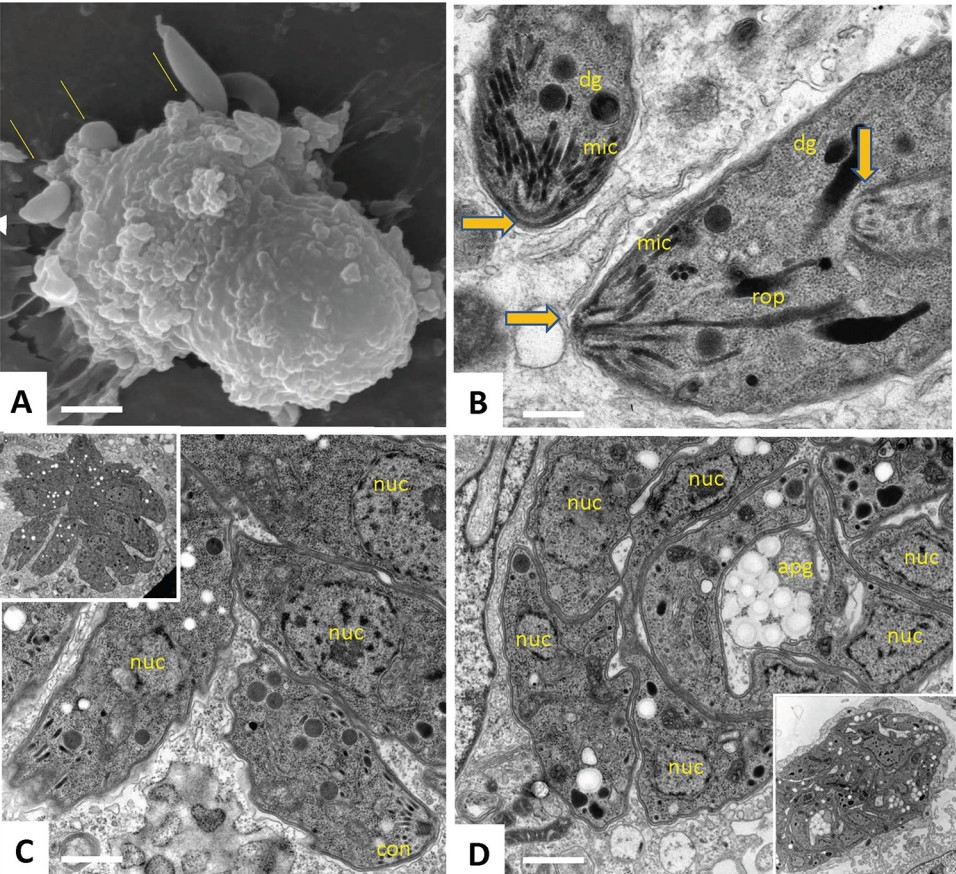

**Fig 4. Scanning and transmission electron microscopy of tachyzoites.** **(A)** Invasion of a single host fibroblast *in vitro* by multiple tachyzoites derived from tissue cysts (thin arrows). **(B)** Tachyzoite dividing by endodyogeny. Thicker arrows point towards the conoids. **(C)** Tachyzoites undergoing proliferation inside the parasitophorous vacuole forming a rosette (insert provides an overview at lower magnification). **(D)** Tightly packed tachyzoites within a parasitophorous vacuole (insert shows lower magnification view). apg: amylopectin granules, con: conoid, dg: dense granules, mic: micronemes, nuc: nuclei, rop: rhoptries. Bars in A = 2 μm, B = 0.3 μm, C and D = 0.6 μm.

Several *B. besnoiti* Lisbon 14 strain tachyzoites can invade a single host fibroblast (Fig 4A), and several parasitophorous vacuoles can be formed. Tachyzoites, exhibiting essentially a similar ultrastructural organization as bradyzoites, divide by endodyogeny (Fig 4B), and often form rosettes, as indicated in Fig 4C. In larger vacuoles, tachyzoites appear very tightly packed, leaving little space for a matrix, and amylopectin granules can also be found in tachyzoites that occupy large vacuoles (Fig 4D) in contrast to *T. gondii* where amylopectin granules are a hallmark of bradyzoites with only few [17] or none [16] present in tachyzoites.

## The genomic sequence of the *Besnoitia besnoiti* Lisbon 14 strain

For whole genome sequencing, we used a clonal line derived from parasites harvested from an infected cow in Lisbon, Portugal. Tissue cysts collected from this cow were mechanically ruptured to release bradyzoites and used to infect cultured human foreskin fibroblasts. The bradyzoites differentiated to tachyzoites which formed large PVs in host cells and were subsequently propagated *in vitro* by serial passaging. We used single molecule real time sequencing (SMRT) to obtain raw genome sequence data from which contaminating host sequences (*Homo sapiens*) were removed. Incorporation of structural variation and single nucleotide polymorphisms

(SNPs) into the previously published Ger-1 assembly and polishing yielded 186 contigs and a total genome size of 58'846'620 bp. The total genome size was estimated at 58.9 Mb. This is in the same range as for the phylogenetically related coccidia *N. caninum* (57.5 Mb), *T. gondii* ME49 (65.7 Mb) and *Eimeria tenella* (51.9 Mb) (Table 1). The median contig length (N50) is 4,079,903 bases.

To build a reference sequence of *B. besnoiti* Lisbon 14 strain (BbLis14), we used the assembly of *B. besnoiti* strain Ger1 [18] as a scaffold and incorporated homozygous sequence variants from DNA-Seq (PacBio) from tachyzoites and RNA-Seq (Illumina) from tachyzoites and tissue cysts. The final BbLis14 reference sequence contained 969 deletions (total length of 11,058 bp), 2,363 insertions (total length of 11,750 bp), and 2,391 SNPs. Using RepeatModeler, we identified 75 long interspersed nuclear elements (LINEs)/Dualen, 70 short interspersed nuclear elements (SINEs), and 2,138 DNA/CMC-EnSpm transposable elements and 16,845 simple, 1,808 low complexity, 1,720 rRNA, and 5,349 unknown repeats. Complementing this, we also used RepeatMasker and the *Sarcocystidae* annotation (almost exclusively data from *T. gondii*) available from RepBase to identify repeats and potential transposable elements. In total, we could identify 97 rRNA, 1839 low complexity, and 18,794 simple repeats.

To annotate the genome, we first transferred the annotation from Ger-1 and then searched for additional genes with Braker2 using the RNA-Seq data and the proteins from *T. gondii*. Homologues were identified through sequence similarity searches to the proteomes of *T. gondii*, *N. caninum*, *Plasmodium falciparum*, and *E. tenella*. In total 8,491 genes were identified with this workflow (S1 Table). Some *T. gondii* genes have several *B. besnoiti* homologues e.g., cytochrome b (TGME49_330000) or cytochrome c oxidase, subunit III (TGME49_323400, S1 Table) which are known *T. gondii* mitochondrial genes with pseudogenes in the nuclear genome. Out of the 8,491 genes, 8,059 were shown to be expressed in tachyzoites and/or bradyzoites; for 432 genes, no significant expression was detected in both developmental stages. Among these non-expressed genes, many are annotated as cytochromes and a few as ribosomal proteins. Many of the non-expressed genes are predicted to code for very small proteins which may be a result of gene fragmentation, an incorrect gene model, annotation or pseudogenes. A majority of the 432 genes with this profile, comprise genes either without homologues in *T. gondii* or with homologues that are not expressed in any *T. gondii* life-cycle stage (S2 Table). Another fraction of these 432 genes (corresponding to 5.1% of the protein-coding genes) contains genes that code for cat-stage or oocyst proteins in *T. gondii*, i.e. stages that currently cannot be queried directly in *B. besnoiti*. In comparison, 1051 out of 8322 genes (12.6%)

**Table 1. Comparison of genome parameters from different Coccidia.**

|  | *Besnoitia besnoitia* Lisbon14 | *Besnoitia besnoitia* Ger1 [18] | *Toxoplasma gondii* ME49 [19] | *Neospora caninum* Liverpool [20] | *Eimeria tenella* Houghton [21] |
|---|---|---|---|---|---|
| **Genome size (Mb)** | 58.8 | 58.9 | 64.9 | 61.5* | 53.3* |
| **Chromosomes** | n.a. | 13* | 13 | 13 | 15 |
| **Coverage (x)** | 30 | 81 | 105 | >100 | 41 |
| **Contigs/scaffolds** | 168 | 172* | 38 | 31 | 35/33 |
| **Median contig N50 (Mb)** | 4.1 | 4.0 | 6.7 | 6.4 | 14 |
| **G+C content (%)** | 57.0 | n.a. | 52.4 | 54.8 | 58.6 |
| **Protein coding genes** | 8491 | 8224* | 8322* | 7364* | 7628* |
| **Accession** | SRA: PRJNA386239 | NCBI BioProject: PRJNA544713 | GenBank: PRJNA638608 | NCBI BioProject: PRJNA531306 | ENA: PRJEB43184 |

* ToxoDB release 55

are not expressed either in *T. gondii* tachyzoites or bradyzoites. For 267 *B. besnoiti* genes, no Ger1 ID could be assigned out of which only 12 genes show no expression (S1 Table). Taken together, we confirmed expression of most previously annotated *B. besnoiti* genes and extended the genome resources by identifying 267 novel transcripts.

## Synteny and orthology between the cyst forming *B. besnoiti*, *T. gondii*, *N. caninum*, and *E. tenella*, and *P. falciparum*

Synteny analysis between *B. besnoiti*, *T. gondii* and *N. caninum* was performed with MCScanX using pairwise determination of syntenic regions inferred from the order of orthologues. Contigs of all three assemblies were arranged according to syntenic relationships and plotted. Contigs shorter than 100 kb were removed for plotting (Fig 5A). Recent re-sequencing of various *T. gondii* strains and *N. caninum* showed chromosome VIIb and VIII to be fused [19, 20] which we have considered for *N. caninum*.

Synteny preservation is high and shows genomic arrangement of the *B. besnoiti* contigs into chromosomes very similar to the *N. caninum* and *T. gondii* genomes with distinct re-arrangements. Chromosomes XII from *N. caninum* and chromosome Ib from *T. gondii* show the highest degree of synteny (Fig 5A). *B. besnoiti* chromosome XI is syntenic to a continuous fragment of *T. gondii* chromosome IX, but synteny breaks occur in the comparison with *N. caninum* for which two fragments are syntenic to parts of *N. caninum* chromosomes V and II. All other *B. besnoiti* chromosomes display several synteny breaks (Fig 5A), but such breaks also occur between *N. caninum* and *T. gondii* (Fig 5B). However, it is to be considered that the

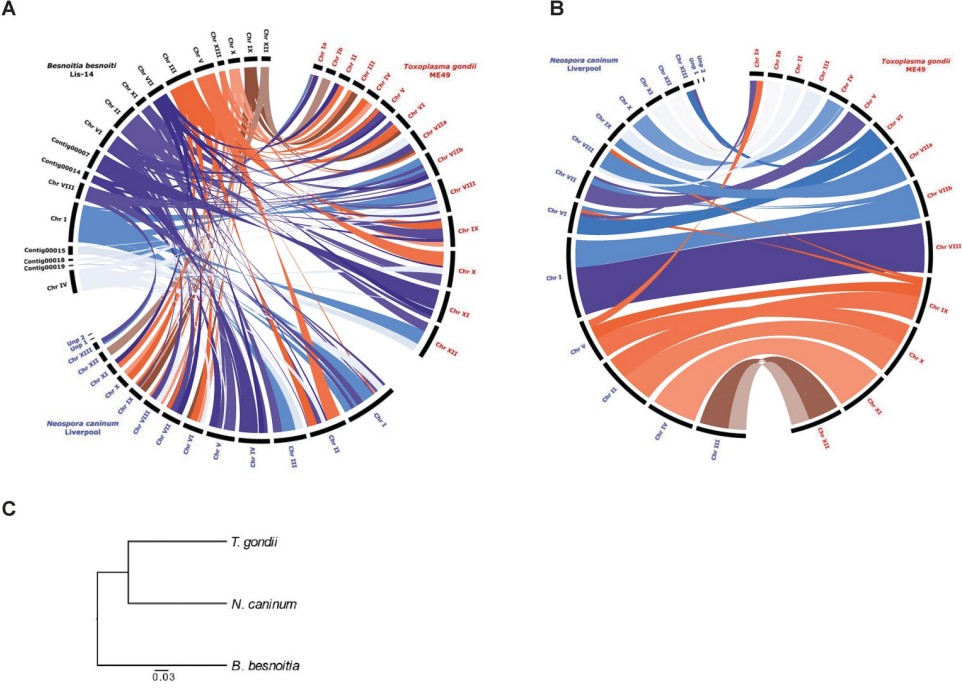

**Fig 5. Synteny and orthology between *Besnoitia besnoiti*, *Toxoplasma gondii* and *Neospora caninum*. (A)** Synteny mapping of the 19 contigs of *B. besnoiti* that are larger than 100 kb shows the genomic organization of the *B. besnoiti* genome to *T. gondii* and *N. caninum*. Syntenic regions are highlighted with coloured lines within the circle. Labelling of the *B. besnoiti* chromosomes is according to GenBank (*B. besnoiti* Ger1 reference strain). **(B)** Synteny mapping as in (A), but exclusively of *T. gondii* and *N. caninum* contigs. **(C)** Phylogenetic organization of *B. besnoiti*, *T. gondii* and *N. caninum*. Branch length represents substitution rate (substitutions per site). Chr: Chromosome; Unp: Unplaced contig.

synteny models presented here do not yet include the fusion of the *T. gondii* chromosomes VIIb and VIII which was established only very recently. Several medium to smaller sized contigs in *B. besnoiti* have not been assigned to chromosomes yet (Fig 5A). Nevertheless, it is evident that the genomes of *N. caninum* and *T. gondii* have a higher degree of synteny to each other than either one has to *B. besnoiti*, which is in line with the greater phylogenetic distance to both genera [22] (Fig 5C).

For global comparison of *B. besnoiti* genes with other apicomplexan parasites, we performed an orthogroup analysis with 3 tissue cyst-forming coccidia (*T. gondii*, *N. caninum* and *B. besnoiti*). *E. tenella* that does not infect the host chronically and is monoxenous, as well as *P. falciparum*, an apicomplexan parasite that is not a coccidian, were used as outgroups (Fig 6 and Table 2). We found 5615 orthogroups comprising a total of 9110 genes that contained at least one gene of each of the cyst-forming coccidia (S3 Table). Interestingly, orthogroup analysis identified 2085 groups containing a single unique *B. besnoiti* gene each or groups with exclusively *B. besnoiti* genes. Among these we find a high number of cytochromes that do not seem to be expressed in tachyzoites or tissue cysts. In addition, many of these unique genes code for surface proteins such as SAG-related surface proteins (SRS). Of these 2085 genes unique to *B. besnoiti*, 720 cannot be annotated using Pannzer or an alternative orthology search algorithm on ToxoDB (S3 Table). Some of these gene products may be important for the interaction of *B. besnoiti* with its specific intermediate hosts and its yet unknown definitive host. Indeed, among these genes, many are not expressed in tachyzoites or tissue cysts and may be important in life cycle stages in the definitive host of *B. besnoiti*.

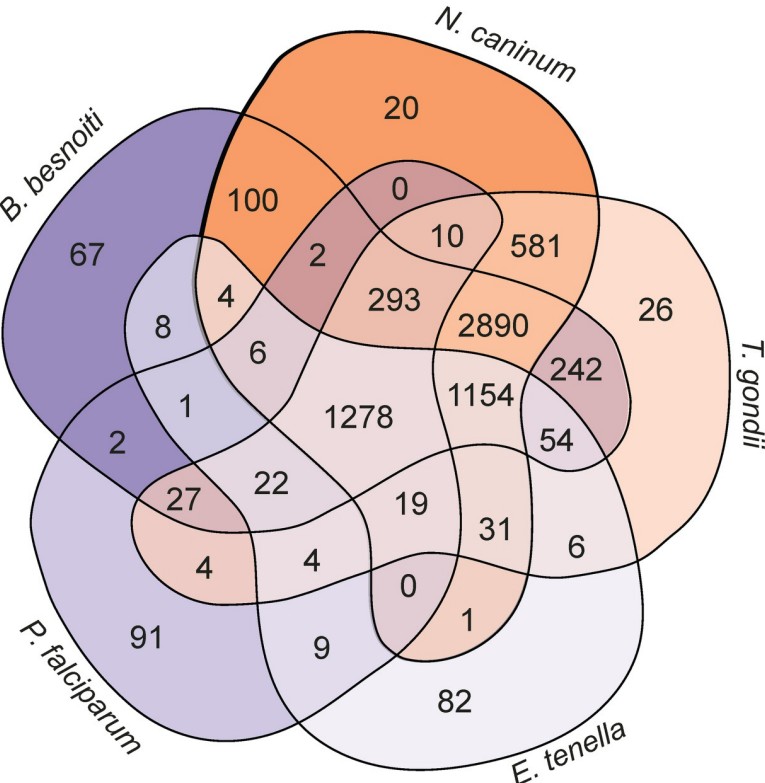

**Fig 6. Orthology between species and unique *B. besnoiti* proteins.** Venn diagram with orthologous groups in *B. besnoiti* and four other apicomplexan species.

In the absence of an identified carnivorous/scavenger definitive host, it can be hypothesized that *B. besnoiti* may undergo sexual development in a blood-feeding insect host, similar to *Plasmodium*. We have therefore compared orthogroups that are common only between *B. besnoiti* and *P. falciparum* and analyzed the expression values of *P. falciparum* asexual blood stages, oocysts and sporozoites (S3 Table). Many orthogroups comprise only ribosomal proteins or mitochondrial components. Although they may be stage-specifically regulated, they are not specialized genes for parasite development in insects. Very few orthogroups were identified that show expression maxima in *P. falciparum* oocysts or sporozoites but are not expressed in *B. besnoiti* as would be expected for genes with roles in a putative *B. besnoiti* sexual cycle in insects, indicating that definitive hosts are unlikely to be insects.

## Differential analysis of mRNA expression between tachyzoites and tissue cysts reveals a large set of stage-specific genes and a conservation of marker genes in tissue cyst-forming coccidia

For the generation of the tachyzoite transcriptome, we harvested tachyzoites cultivated *in vitro* in human foreskin fibroblasts and extracted total RNA from parasite-enriched fractions. Tissue cysts were isolated directly from the dermis of an infected cow into PBS and pools of cysts were subjected to extraction of total RNA. Quality control using the Agilent Tapestation confirmed a high proportion of parasite-specific RNA in both preparations (S1A Fig). Total RNA was enriched for mRNA and subjected to RNA-Seq. Read mapping was done using the predicted *B. besnoiti* Lis14 cDNA sequences yielding a total of 271'210'209 reads for tachyzoites (247'345'532 unique mapped reads, 91.20%) and 289'906'951 total reads for tissue cysts (243'282'058 unique mapped reads, 83.92%, S1B Fig). Highly regulated genes for both stages (LFC >|3|) are displayed in a heatmap. Using a stringent 4-fold regulation cut-off, 897 and 1146 genes are significantly higher expressed in tachyzoites and tissue cysts, respectively (Fig 7A and S4 Table). Canonical markers for tachyzoites or bradyzoites in *T. gondii* are plotted and show the expected stage-specificity of RNA expression except for microneme protein 13 (MIC13) [23–25], lactate dehydrogenase (LDH) 1 [26] and matrix antigen (MAG) 1 [27] that are not developmentally regulated in *B. besnoiti* (Fig 7B and 7C). SRS44 (TgCST1) is a major structural protein of the tissue cyst wall in *T. gondii* containing a mucin domain which binds the lectin *Dolichos biflorus* agglutinin (DBA) and used as a marker for *T. gondii* tissue cyst walls in immunofluorescence assays (IFA) [28]. The closest *B. besnoiti* homologue, BESB_048010, is highly upregulated (> 240-fold) in tissue cysts. The lack of a protocol to induce stage-differentiation and generate *B. besnoiti* tissue cysts in *vitro*, prevents direct validation of this gene as encoding a DBA-binding mucin. In addition, the predicted gene model (S2A and S2B Fig) is not fully supported by RNA-Seq data: The RNA-seq data suggest that intron 2 predicted in the gene model is the only spliced intron in the pre-mRNA, whilst the predicted introns 1 and 3 are represented by transcribed reads (S2A Fig) and therefore remain part of the mature mRNA. To validate this, we performed rapid amplification of cDNA 3' ends

**Table 2. Number of genes in orthogroups.**

| Species | Total number of genes | Genes in 1 species | Genes in 2 species | Genes in 3 species | Genes in 4 species | Genes in 5 species |
|---|---|---|---|---|---|---|
| *B. besnoiti* | 8491 | 2081 | 416 | 3176 | 1507 | 1311 |
| *N. caninum* | 7364 | 707 | 747 | 3120 | 1490 | 1300 |
| *T. gondii* | 8322 | 1506 | 894 | 3099 | 1512 | 1311 |
| *E. tenella* | 8597 | 5923 | 24 | 97 | 1230 | 1323 |
| *P. falciparum* | 5389 | 3664 | 15 | 52 | 345 | 1313 |

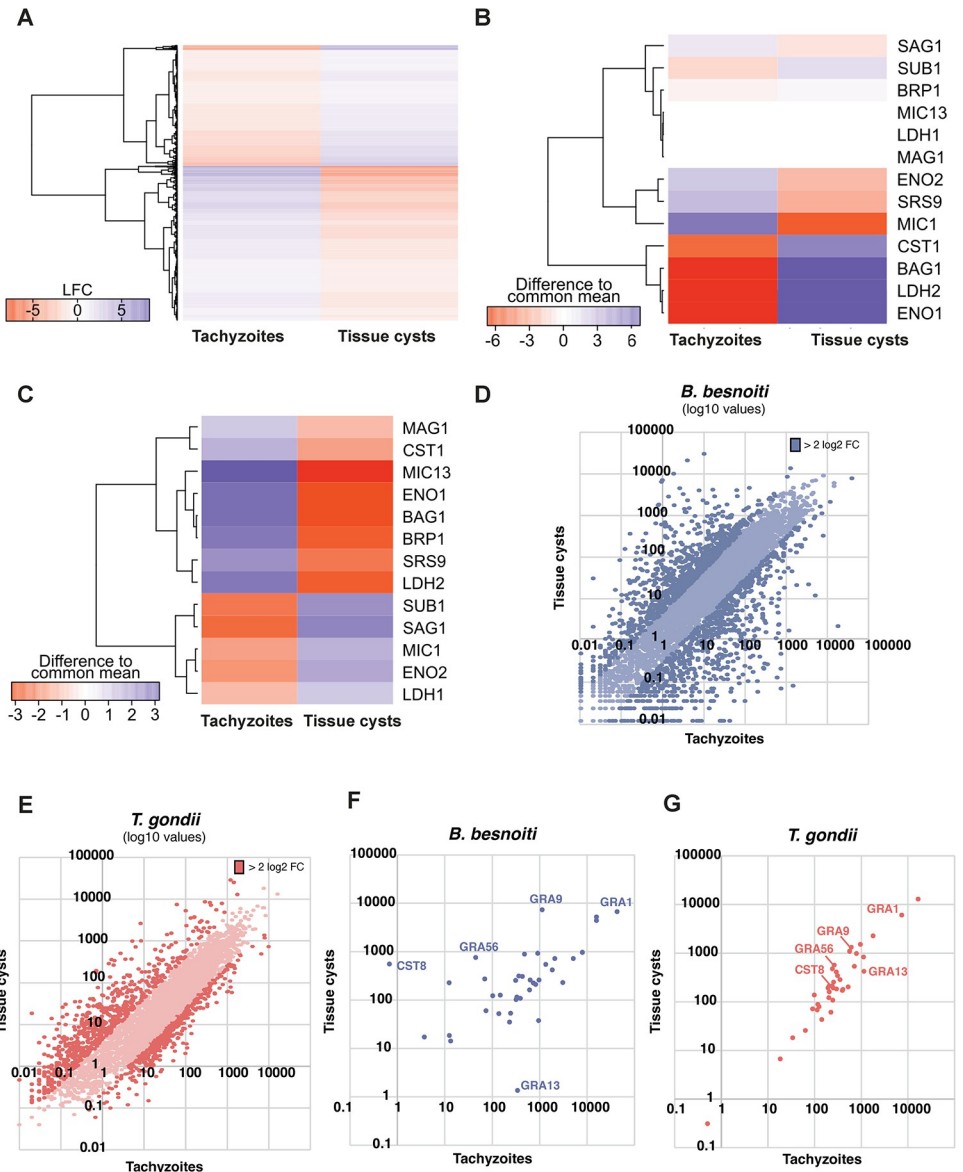

**Fig 7. Differential expression of mRNA between *B. besnoiti* tachyzoites and tissue cysts and comparison to *T. gondii*.** **(A)** Heatmap depicting log$_2$ fold changes (LFC) of the top 10% of differentially expressed genes. N = 1172 **(B)** Heatmap showing the differences to the common mean (Z-score) of *B. besnoiti* orthologues of typical stage marker genes known in *T. gondii*. Tachyzoites (Z < 0, red) and tissue cysts (Z > 0, purple). **(C)** Heatmap showing the differences to the common mean (Z-score) of *T. gondii* tissue cyst-specific genes. **(D)** Scatter plot of TPM (Transcripts Per Kilobase Million, ToxoDB release 52) values of expressed genes in *B. besnoiti*. Values for tachyzoites are plotted against TPMs from tissue cysts. **(E)** Scatter plot of TPMs (ToxoDB release 52) values of protein coding genes in *T. gondii*. **(F)** Scatter plot with TPMs depicting genes encoding expressed dense granule proteins in *B. besnoiti*. **(G)** Scatter plot with TPMs depicting syntenic *T. gondii* genes from figure (F). BAG1: bradyzoite antigen 1; BPK1: bradyzoite pseudokinase 1; BRP1: bradyzoite rhoptry protein 1; CST: cyst wall protein; ENO: enolase; LDH: lactate dehydrogenase; MAG: matrix antigen; MIC: micronemal protein; SAG: surface antigen; SRS: SAG-related sequence; SUB: subtilisin.

(RACE) PCR. We obtained 3 amplicons of different sizes from BESB_048010 cDNA. Sanger sequencing of these PCR products revealed 3' UTR polymorphisms (S2B Fig). These analyses also showed that the predicted intron 3 is not spliced. A PCR product encompassing the predicted coding sequence (CDS) of BESB_048010 yielded 2 bands > 7 kb. Sequencing by primer

walking confirmed the absence of intron 3 as well as intron 1, whilst the predicted intron 2 appeared to be removed by splicing. We detected 2 shorter transcript isoforms with alternatively spliced products (S2B Fig). Sequence analysis predicts that the functional start codon is 159 bp downstream of the translation start predicted in the original gene model. These newly described transcripts code for predicted proteins containing several SRS domains (S2C Fig). In *T. gondii*, the CST1 primary structure predicts a signal peptide and 9 SRS domains, and an extensive stretch containing threonine-repeats interspersed with lysine, proline and arginine residues. Threonine and hydroxyproline can be O-glycosylated, and regions enriched for these amino acids are a hallmark of mucin domains. Taken together, all *B. besnoiti* predicted SRS44 orthologues validated by RT-PCR sequencing lack a canonical signal peptide and importantly, a putative mucin domain (S2C Fig) and therefore, based on the prediction of the primary open reading frame, there is currently no evidence for this protein being a target for DBA.

*B. besnoiti* orthologues of 20 genes coding for factors which have been localized to the tissue cyst wall in *T. gondii* (S5 Table) were identified, and 5 genes show at least 4-fold upregulation in tissue cysts, with the majority being significantly expressed albeit not stage-regulated in *B. besnoiti*. Not unexpectedly, the genetic underpinnings of tissue cyst formation show considerable species-specific differences between *T. gondii* and *B. besnoiti*. Orthologues for 10 of the 31 listed *T. gondii* cyst wall proteins are not represented in the *B. besnoiti* genome, which may be a consequence of development of *B. besnoiti* tissue cysts in the dermis versus brain and muscle for *T. gondii* and *N. caninum* with host and parasite species-specific adaptations of cyst wall structures.

Comparing gene expression between tachyzoites and bradyzoites from isolated tissue cysts in *B. besnoiti* and *T. gondii* shows that more genes appear differentially expressed in *B. besnoiti* as illustrated by the higher dispersion from the diagonal (4019 in *B. besnoiti* versus 3557 in *T. gondii* with a fold change > 2, Fig 7D and 7E), with a similar number of genes analyzed (see Table 1). This may be partially explained by a background level of bradyzoite-specific gene expression in cultured *T. gondii* tachyzoites whilst we did not observe this for *B. besnoiti* Lis14 tachyzoites. This is consistent with *B. besnoiti* being completely refractory to induction of encystation by stress conditions *in vitro* which are used to drive *T. gondii* into differentiation to tissue cysts. This favorable signal-to-noise ratio in *B. besnoiti* is exemplified by putative tissue cyst-specific GRA transcripts (Fig 7F and 7G and S6 Table): Whilst GRA13 mRNA is more abundant in *B. besnoiti* tachyzoites compared with bradyzoites, it is not stage-regulated in *T. gondii*. Similarly, GRA9, GRA56 and CST8/GRA54 are upregulated in *B. besnoiti* bradyzoites, but to a lesser extent (GRA9, GRA56) or not at all (CST8) in *T. gondii* bradyzoites (S6 Table and Fig 7F and 7G). Several *T. gondii* genes coding for dense granule proteins have no orthologues in the *B. besnoiti* genome (S6 Table) which may again indicate adaptation to the host cell niche. It is highly probable, that several *B. besnoiti* dense granule proteins are yet to be identified. More detailed analysis of key secreted protein families, the GRAs, MICs, ROPs, and GPI-anchored SRSs with orthologues in *T. gondii* show a similar diversity of annotated family members (S7 Table). A graphical depiction of differential expression of these conserved proteins (S3 Fig) representing normalized read counts displayed in S7 Table also illustrates that a majority of these family members are not stage-specifically expressed.

In contrast to *T. gondii* cysts, *B. besnoiti* tissue cysts isolated from the dermis of an infected bovine can be prepared to considerable purity and contain very large numbers of bradyzoites, rendering this parasite highly attractive to study its molecular composition. Species-specific regulated genes may provide direct leads for the investigation of differences in structure, for example a prominent, presumably parasite-induced, and host-derived layer of collagen around the host cell harboring the tissue cyst, as well as factors determining tissue tropism of cysts. Proteomic analyses of whole tissue cysts and isolated cyst walls may identify molecules

especially important in the maintenance and survival of *in vivo* cysts in *B. besnoiti*, in particular for all tissue CFA. To date, there is no procedure for inducing tissue cyst formation *in vitro* which precludes systematic molecular genetic investigations of this life cycle stage. The considerable similarity of the cyst-specific genetic program of *T. gondii* and *B. besnoiti* despite the clear differences in structural elements, size, cell and tissue tropism, and morphology provides exciting opportunities for comparative evolutionary biology.

## *In silico* metabolic analysis of *B. besnoiti* tachyzoites and tissue cysts

As the acute tachyzoite stage proliferates rapidly, high energy turnover is necessary. It is therefore thought that the parasite, along with the uptake of nutrients from its host niche, exploits its *de novo* synthesis machineries. Using functional annotation, we identified 59 metabolic pathways with 692 metabolic genes for *de novo* biosynthesis of essential metabolites or its acquisition from the host (Fig 8A and Tab A and Tab B in S8 Table). These genes were

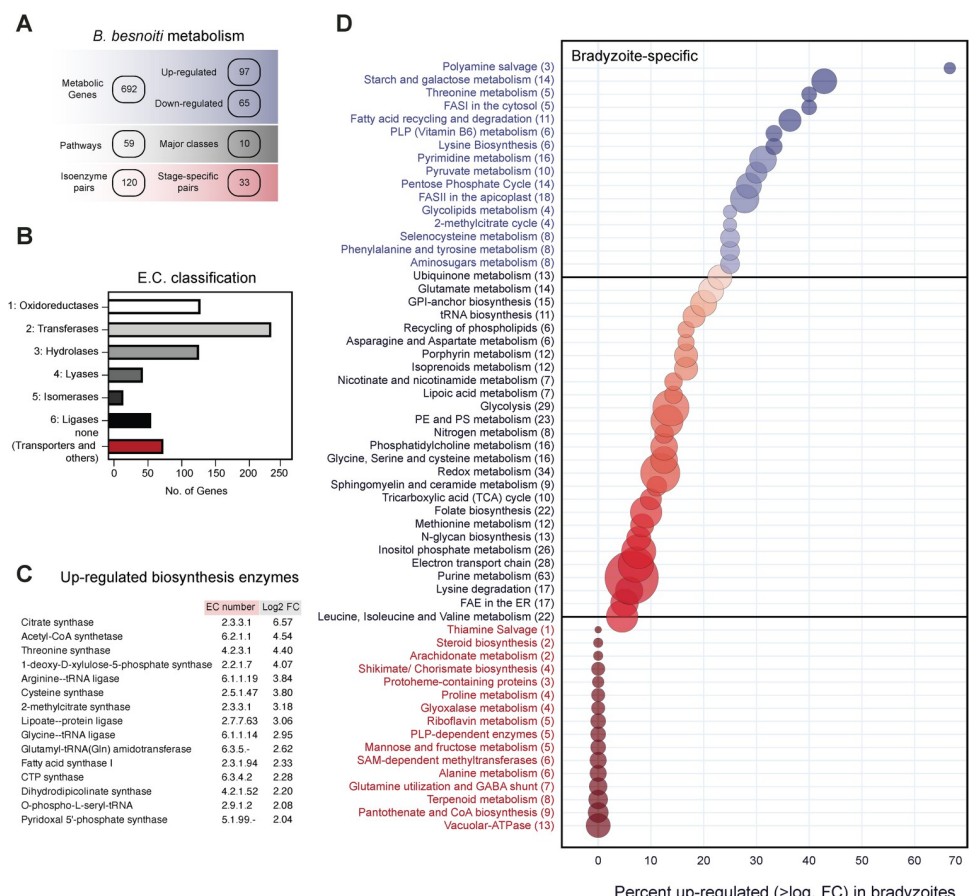

**Fig 8. Metabolic pathways of *B. besnoiti*. (A)** Overview and classification of metabolic genes: number of annotated genes, pathways, and isoenzymes in *B. besnoiti*. Up- and downregulated genes have a relative fold-change >4 ($\log_2 > |2|$). Stage-specific isoenzymes are pairs that display upregulation only in one stage of the parasite (bradyzoites or tachyzoites; see also Tab E in S8 Table). **(B)** E.C. classification of all enzymes found in the *B. besnoiti* genome. Transferases are the most common class of enzymes, followed by transporters and enzymes with an unknown function (see also Tab F in S8 Table). **(C)** List of up-regulated genes encoding biosynthesis enzymes by the cyst-forming *B. besnoiti* bradyzoites. These enzymes are part of diverse metabolic pathways, including several tRNA-synthetases. **(D)** The 59 curated metabolic pathways identified in *B. besnoiti*, ordered by percentage of genes upregulated in the bradyzoite stage. Bubble size represents the number of genes (in brackets) in each pathway. Eighteen pathways have more than 25 percent of their genes upregulated in the bradyzoite stage.

categorized based on their enzyme classification (EC) numbers. More than 200 genes belonged to the class of transferases (Fig 8B) representing the largest category. Comparing the RNA-Seq data of Lis14 bradyzoites and tachyzoites revealed that several pathways were differentially regulated based on mRNA expression. Of the 692 identified genes, 97 genes were upregulated ($>2$ log$_2$FC) in bradyzoites (Tab C in S8 Table), comprising 14% of all known metabolic genes. Of these 97 genes, 36 (37%) belonged to only eight pathways: polyamine salvage, starch and galactose metabolism, threonine metabolism, fatty acid synthesis (FASI), fatty acid degradation and recycling, pyridoxal-5P (PLP, vitamin B6) biosynthesis, lysine biosynthesis and pyrimidine biosynthesis. Additionally, the transcription of several synthases in vitamine, amino-acid and fatty-acids pathways were also up-regulated in bradyzoites, suggesting a greater utilization of the *de novo* biosynthesis pathway during latency (Fig 8C). All the 59 pathways identified in *B. besnoiti* were ordered according to the percentage of genes in a pathway upregulated in bradyzoites with the pathways containing most of the upregulated genes appearing at the top (Fig 8D). Interestingly, genes encoding apicoplast-resident metabolic proteins are also expressed in *B. besnoiti* bradyzoites, suggesting the utilization of *de novo* synthesized fatty acids via the fatty acid synthesis (FASII) pathway for latency. The FASII pathway has been shown to be fitness conferring in *T. gondii* tachyzoites and supplementation of short or medium chain fatty acids (C14/ C16 FAs) were shown to rescue intracellular growth defects [29–31]; however, the role of this pathway or FA salvage by bradyzoites remains unknown. The two pathways with the most genes upregulated in bradyzoites are the starch and galactose metabolic (Fig 9A) and polyamine salvage (Fig 9B) pathways. The metabolic pathway of starch and galactose is present within the coccidian-specific branch of Apicomplexa and enables the storage of carbohydrates in granules of amylopectin. In *B. besnoiti* bradyzoites, several enzymes are involved in the anabolism of disaccharides or floridean starch (amylopectin variant found in red algae, Fig 9A). It is possible that sugars and starches are important components of the cyst wall and/or starches may mediate long-term storage of sugars. Polyamines, on the other hand, are derived from amino acids and have been shown in single celled eukaryotes to be essential to growth and proliferation (reviewed by Michael [32]). *B. besnoiti* bradyzoites upregulate the expression of all three enzymes involved in the degradation of spermine for spermidine and finally putrescine (Fig 9B) indicating that polyamines play an important, but as-yet unknown role during the tissue cyst stage.

In contrast to *B. besnoiti*, *T. gondii* bradyzoites upregulate genes coding for enzymes involved in starch catabolism whilst tachyzoites upregulate genes for anabolic enzymes involved in the biosynthesis of floridean starch [33]. This is highly intriguing and indicates a very different and possibly more active metabolism of *B. besnoiti* tissue cysts that not only acts as energy stores, but probably also to produce structural components of the cyst wall. The starch and galactose pathway also reveals that trehalose synthesis mediated by the bi-functional trehalose phosphate synthase/trehalose phosphatase is up-regulated in *B. besnoiti* tachyzoites and may be an interesting drug target for acute besnoitiosis [34] due to its absence in the mammalian host.

To identify stage-specifically upregulated pathways, we plotted all the log$_2$ fold changes of the normalized read counts between bradyzoites and tachyzoites in *B. besnoiti* against the values from *T. gondii* (Fig 9C and Tab D in S8 Table) [9]. Canonical bradyzoite-specific genes in *T. gondii* such as enolase 1 (ENO1), lactate dehydrogenase 2 (LDH2) and phosphoenolpyruvate-carboxykinase (PEPCK) are also upregulated in *B. besnoiti* bradyzoites. Similarly, known tachyzoite markers in *T. gondii* such as enolase 2 (ENO2), nucleoside transporter (NT) 1 and 2 display tachyzoite-specificity in *B. besnoiti*. Striking differences between the two species can be seen in the mRNA expression profiles of the sphingolipid biosynthesis gene serine palmitoyltransferase (SPT1), and the glycosylphosphatidylinositol (GPI)-anchor biosynthesis

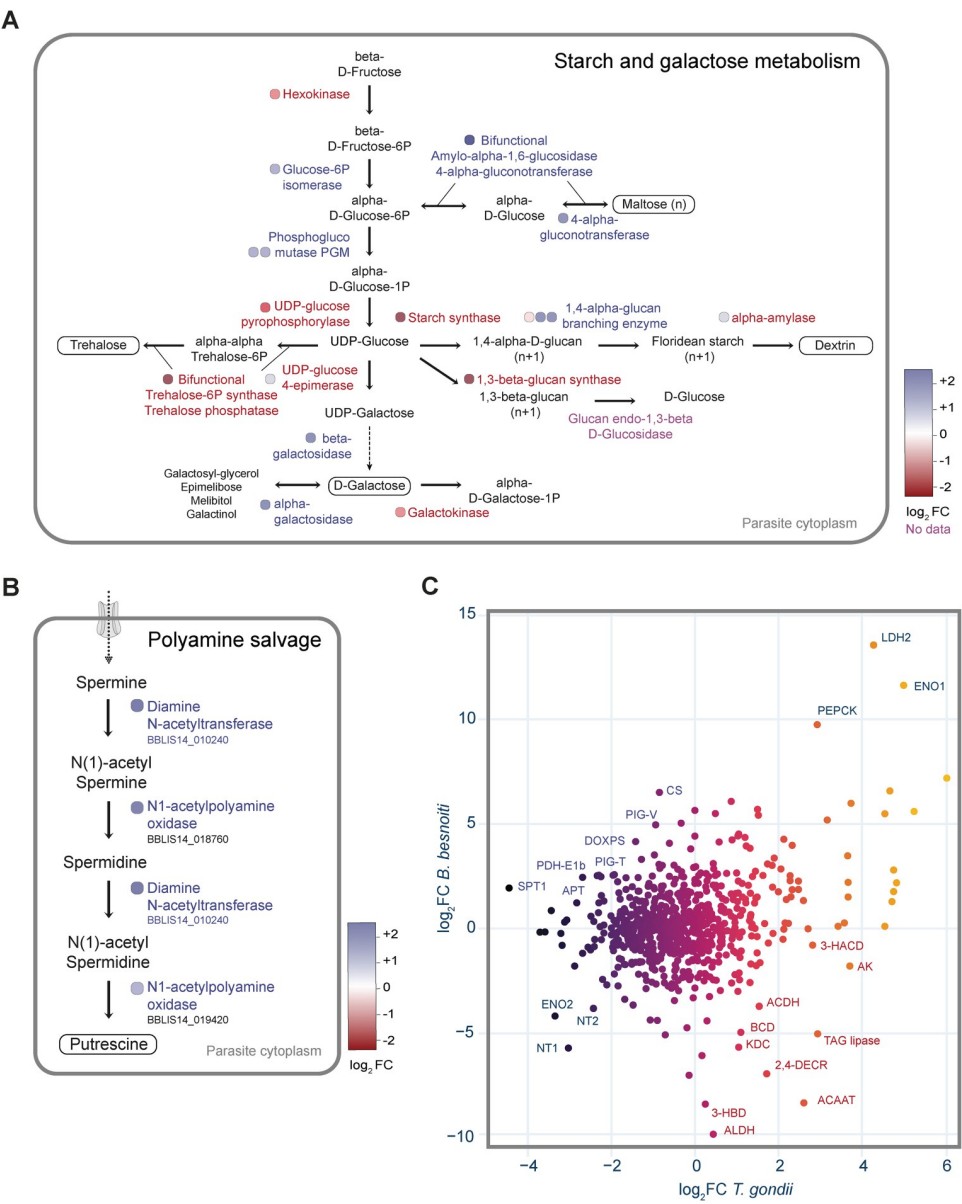

**Fig 9. Upregulated pathways and correlation of gene expression levels the *T. gondii* transcriptomic profile in bradyzoites. (A)** The starch and galactose metabolic pathway with their gene expression in bradyzoites highlighted from red (downregulated) to blue (upregulated). Bubbles in shades of red or blue mark the level of regulation. Note that hexokinase belongs to the glycolysis pathway but is highlighted because its activity initiates the starch and galactose pathway. **(B)** Schematic representation of the polyamine salvage pathway and their high expression level in the latent stage of *B. besnoiti*. Bubbles in shades of blue highlight the high level of upregulation in *B. besnoiti* bradyzoites. **(C)** Correlation of the ratio of gene expression between bradyzoites and tachyzoites in *B. besnoiti* compared to *T. gondii* of all known metabolic genes in these coccidan species. Genes upregulated only in *B. besnoiti* but not in *T. gondii* are highlighted in blue and highlighted in red are genes upregulated only in *T. gondii* bradyzoites. 2,4-DECR: 2,4-dienoyl CoA reductase 2; 3-HACD: 3-hydroxyacyl-CoA dehydrogenase; 3-HBD: 3-hydroxyisobutyrate dehydrogenase; ACAAT: acetyl-CoA C-acyltransferase, ACDH: Acyl-CoA dehydrogenase domain-containing protein; AK: adenylate kinase; ALDH: alanine dehydrogenase; APT: aminopeptidase n; BCD: Butyryl-CoA dehydrogenase; CS: citrate (si)-synthase; DOXPS: 1-deoxy-D-xylulose 5-phosphate synthase; ENO1: enolase 1; ENO2: enolase 2; KDC: lysine decarboxylase family protein; LDH2: lactate dehydrogenase 2; NT1: nitrite transporter 1; NT2: nitrite transporter 2, PDH-E1b: pyruvate dehydrogenase E1 beta subunit; PEPCK: Phosphoenoylpyruvat (PEP) carboxykinase; PIG-T: GPI-anchor transamidase; PIG-V: mannosyltransferase; SPT1: serine C-palmitoyltransferase 1; TAG lipase: triacylglycerol lipase.

genes PIG-V (GPI mannosyltransferase) and PIG-T (GPI transamidase) which are upregulated in *B. besnoiti* bradyzoites but not in in *T. gondii*. On the other hand, the opposite direction of regulation was shown for genes encoding aldehyde dehydrogenase (ALDH), acetyl-CoA-acyl-transferase (ACAAT), 2,4-dienoyl-CoA reductase (2,4-DECR) and triacylglycerinol (TAG) lipase for which *B. besnoiti* displays strong upregulation in tachyzoites whilst tachyzoites of *T. gondii* downregulate these genes. This suggests fundamental differences in the metabolic capabilities of the two coccidian species, and the utilization of pathways.

The analysis of the metabolic enzymes reveals that the expression levels of stage-specific markers is consistent between *B. besnoiti* and *T. gondii* (Fig 9C). It is to be noted, that our analysis is based on the orthologues of the approximately 500 known genes in *T. gondii*. Thus, unknown, or unannotated metabolic genes were not taken into consideration.

## Genomic analysis reveals the genetic basis for an active canonical coccidian life cycle

The only documented transmission routes for *B. besnoiti* between intermediate hosts are by direct physical contact or mediated by insects, and a definitive host has not been identified. Thus, until now, this precluded addressing the key question whether this coccidian apicomplexan could complete sexual development in a definitive host at all or was transmitted solely between intermediate hosts. Generally speaking, sexual development increases genome diversity, allowing the emergence of more virulent or pathogenic strains and the identification of a definitive host would be important to protect intermediate hosts and livestock from besnoitiosis. We hypothesized that a secondary loss of sexual development in *B. besnoiti* would be associated with the deterioration of genomic information [35] coding for specific functions directly associated with gametogenesis, fertilization, and oocyst development and maturation. Both *T. gondii* and *N. caninum* have a documented sexual cycle (reviewed in [36]), although molecular or microscopical data on the development of *N. caninum* in enterocytes of the dog definitive host are lacking. However, in *T. gondii*, extensive bulk RNA-Seq datasets generated at different time points post inoculation after experimental infection of kittens allowed identification of a total of 485 genes whose expression is restricted to enteroepithelial stages (TgEES) in cats, i.e. with no or minimal expression in tachyzoites or bradyzoites, but with maximal expression in TgEES1-5 (TgEES-R) [37]. The *Tg*EES-R gene set was complemented by 127 previously identified genes with higher mRNA expression in unsporulated and/or sporulated oocysts [38] than in all other stages [37] (TgEES-RO, S9 Table). We assessed representation of *B. besnoiti* orthologues within the 612 ORFs of the combined *T. gondii* TgEES-RO gene set (S9 Table) to estimate the genetic capacity for encoding the various structural proteins and machinery of gamete development in the gut epithelium of a putative definitive host, including fertilization as well as formation and maturation of oocysts. Overall, we find substantial representation of *B. besnoiti* orthologues in the TgEES-RO dataset (387 syntenic orthologues (as defined on ToxoDB) of 612 TgEES-RO genes), albeit with a considerable increase in representation in EES5 and oocysts relative to EES1-4 (Fig 10A). *B. besnoiti* orthologues of EES1-4 genes comprise less than 10% whereas more than 90% of the identified orthologues belong to the EES5 and the oocyst group. For comparison, we identified the percentage of *B. besnoiti* genes (LFC > |2|) that are orthologous to upregulated genes in *T. gondii* tissue cysts (n = 814) (Fig 10B). We find 483 unique *B. besnoiti* genes of which approximately 75% are not differentially regulated between tachyzoites and tissue cysts. Of note: EES1-4 genes account for 22% of all TgEES-RO genes (Fig 10C and 10D). However, among the previously defined categories and key genes [37] are: hypothetical and confirmed "oocyst wall" proteins (10 out of these 11 *T. gondii* OWP genes are represented in the *B. besnoiti* genome), "axoneme" proteins (30 of 33

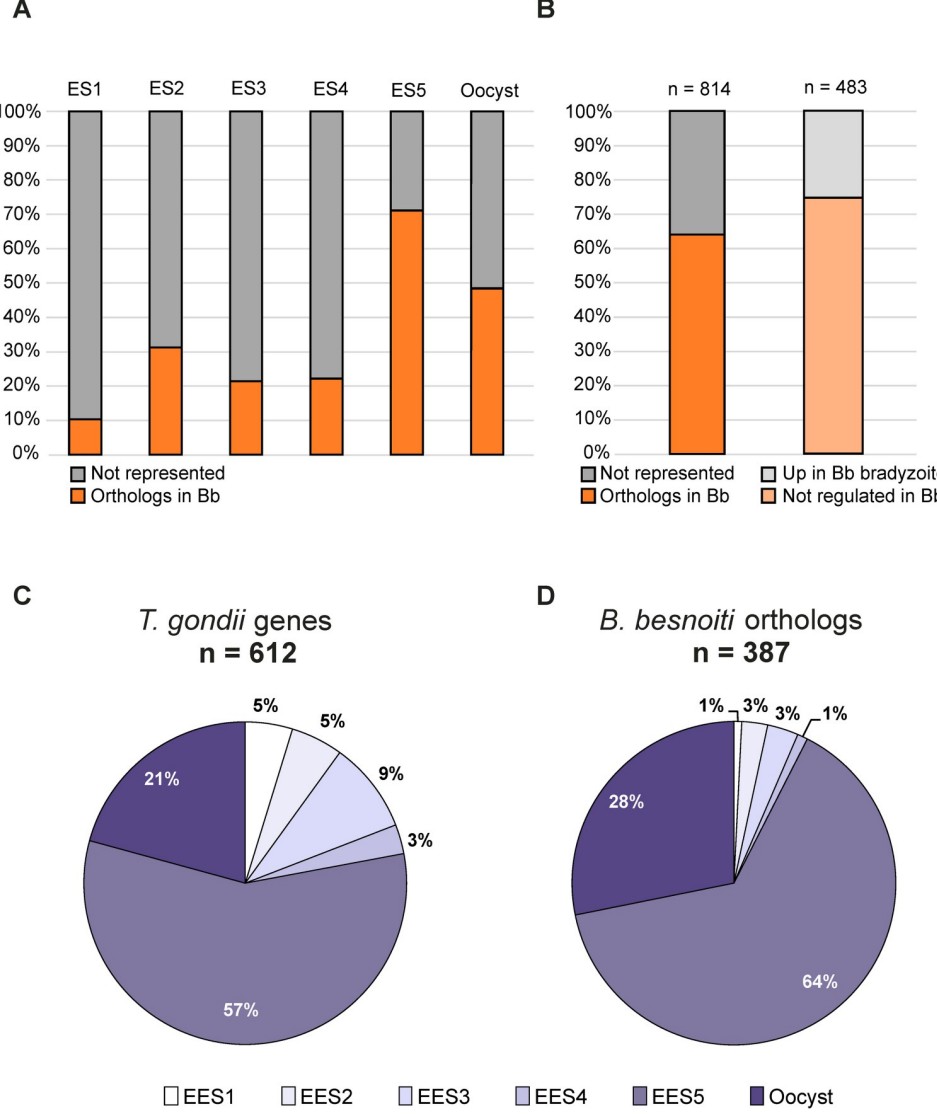

**Fig 10. *B. besnoiti* orthologues involved in sexual reproduction and oocyst formation.** (A) Bar chart showing the percentage of *B. besnoiti* orthologues identified for all *T. gondii* genes with expression maxima EES1-5 or oocysts (oocyst genes listed as in [38]). (B) Percentage of *B. besnoiti* genes (LFC > |2|) identified that are orthologous to upregulated genes in T. gondii tissue cysts (n = 814) (left graph) [38] This results in 483 unique *B. besnoiti* genes of which approximately 75% are not differentially regulated between tachyzoites and tissue cysts (right graph). (C) Percentage of *T. gondii* genes having expression maxima in a distinct coccidian stage. (D) As in (C), but for *B. besnoiti* orthologues.

*T. gondii* genes have orthologues in *B. besnoiti*) and "flagellar transport" proteins (15 of 20 *T. gondii* genes have orthologues in *B. besnoiti*). 265 TgEES-RO genes were conserved in both *B. besnoiti* and *Eimeria tenella* (S9 Table). Of note, a majority of the corresponding *T. gondii* orthologues (257) could be assigned to the gametocyte (EES4-5) and oocyst groups (S9 Table).

The large majority of the 225 TgEES-RO genes which are not represented as orthologues in the *B. besnoiti* genome fall into the categories SRSs (surface antigen 1 (SAG1)-related sequences), Family A-E proteins, and hypothetical genes (S9 Table). Of note, a conserved *B. besnoiti* orthologue coding for the centrally important microgamete fusion factor TgHAP2 (TGME49_285940) necessary for zygote formation was identified (BESB_068050, S4 and S5

Figs). Analysis of sequence alignment (S5A Fig), functional domains (S5B Fig), as well as the predicted 3D structures with iTASSER (S5C Fig) show a high degree of conservation supporting a biologically active role as a key membrane factor for zygote formation in *B. besnoiti*. The iTASSER analysis returned highly similar structure models for the orthologues of both species even in the part which has a lower degree of sequence conservation (S4 Fig). Orthologues of *T. gondii* genes involved in sexual development are not only present but also appear functional based on the lack of significant accumulation of polymorphisms indicating conservation of protein coding sequence.

Stage-specific gene expression and cell cycle regulation in Apicomplexa have been shown to be controlled by apetala 2 (AP2) transcription factors, many of which are expressed stage-specifically. Of the 67 *T. gondii* AP2 (TgAP2) factors annotated in ToxoDB, 15 have no syntenic orthologue in the *B. besnoiti* genome (S10 Table). Importantly, *B. besnoiti* orthologues for 10 of the 12 EES-specific TgAP2s were identified (S10 Table), which is an additional indicator for conservation of the complex regulatory processes governing development in the definitive host. A microorchidia gene repressor (MORC, TGME49_305340) in conjunction with 10 primary AP2s and HDAC3 regulate gene expression on an epigenetic level during sexual development in *T. gondii* [39, 40]. The product of a MORC orthologue in *B. besnoiti* (BESB_017620) is predicted to have KELCH domains as well as an HATPase domain. All but two primary AP2s as well as HDAC3 and additional MORC partner proteins containing ELM2 and PHD domains have orthologues in the B. besnoiti genome (S10 Table). Hence, apart from the *B. besnoiti* orthologues for TgAP2III-1 and TgAP2III-3 all previously identified MORC partners [40] are represented. In addition, 47 MORC-regulated microgamete genes have been identified in *T. gondii* by Farhat *et al.* [40] and subdivided into 3 categories: axoneme, flagellar transport, and "other". Consistent with the presence of genes coding for a MORC orthologue and its partners in *B. besnoiti*, we show that 45 of the 47 listed MORC effector genes in these functional categories are conserved between *B. besnoiti* and *T. gondii* (S11 Table). Taken together, there is robust evidence on the genomic level for an active sexual cycle in *B. besnoiti* in a yet unknown definitive host.

Taken together, this is the first study of *B. besnoiti* combining microscopical analyses, genomics and transcriptomics of two distinct endpoint developmental stages in the intermediate cattle host. Ultrastructure of *B. besnoiti* tachyzoites and tissue cysts demonstrate that organellar structures in zoites are highly similar between *T. gondii* and *B. besnoiti*, and that the tissue cyst wall of the latter is strikingly different, consistent with its niche in the skin. Analysis of the genome strongly suggests that *B. besnoiti* can undergo sexual development and oocyst formation in a definitive carnivore or scavenger host. The *B. besnoiti* transcriptomes of tachyzoites and tissue cysts display a large number of differentially expressed genes with many conserved stage-specific markers in *T. gondii* and *B. besnoiti*. Combined with the recent technical advances, i.e., CRISPR/Cas9 reverse genetics in *B. besnoiti*[41], this study opens up the field to new investigations of this understudied parasite.

## Materials and methods

### Ethics statement

For this study, no experimental infections were performed and thus, no ethics statement is required.

### Isolation and cultivation of *B. besnoiti*

Tissue cysts were isolated from subcutaneous tissues of infected cows that were slaughtered due to advanced besnoitiosis by scraping the tissue material into a Petri dish containing PBS using a syringe needle. Individual cysts were collected with a pipette. Bradyzoites were released

from the cysts by mechanical disruption of the cyst wall with a pestle followed by centrifugation to separate the bradyzoites from the cyst wall. Human foreskin fibroblasts (HFFs) cultivated in Dulbecco's Modified Eagle Medium (DMEM) with GlutaMax (Gibco; Invitrogen; Thermo Fisher Scientific, Inc., Waltham, MA, USA) and 10% heat inactivated fetal bovine serum (FBS) (FBS; Gibco; Invitrogen; Thermo Fisher Scientific, Inc., Waltham, MA, USA) and antibiotics (100 units/ml penicillin, 100 ng/ml streptomycin and 250 ng/ml amphotericin B) were infected with bradyzoites to establish tachyzoite cultures. Limiting dilution cloning was performed in 2 steps to obtain a clonal line. The resulting *B. besnoiti* line was then maintained as tachyzoite culture by serial passage in HFFs in DMEM (Sigma-Aldrich D6429) with high glucose and additional 2 mM L-glutamine (Sigma-Aldrich), 10% FBS and optionally supplemented with 100 units/ml penicillin, 100 ng/ml streptomycin and 250 ng/ml amphotericin B. For passaging, infected host cells were dislodged with a cell scraper and tachyzoites were extracted by syringe homogenization with an 18G and subsequently 22G needle. Free tachyzoites were used to inoculate new host cells. Parasites are cultivated at 37˚C, 5% $CO_2$ and >90% humidity.

## Microscopy of tissue cysts

Tissue cysts were processed for light and electron microscopy (EM) by placing them into primary fixative (2.5% glutaraldehyde in 100 mM sodium cacodylate buffer pH 7.3) for 2 h, followed by washing in cacodylate buffer and post-fixation in 2% osmium tetroxide in cacodylate buffer for 4 h at room temperature. Specimens were washed in distilled water, and sequentially dehydrated in 30, 50, 70, 90, and 3 x 100% ethanol, 5 min each. For SEM, samples were incubated in hexamethyl-disilazane, 2 times 5 min to remove the ethanol, placed onto glass coverslips and allowed to dry overnight. Subsequently, samples were sputter-coated with gold and viewed on a Jeol 840 scanning electron microscope at 25 kV. For sectioning, samples were embedded in Epon 814 epoxy resin and polymerized at 60˚C, and sections were cut on a Reichert and Jung (Vienna) microtome. Conventional sections of 1–2 μm thickness were prepared for visualizing cysts by light microscopy. Following placement of sections on a glass-coverslips, specimens were stained with methylene blue and fuchsin red, washed in water and airdried. For transmission EM, sections of 60–80 nm thickness were placed onto formvar-coated nickel grids and stained with uranyless and lead citrate (Electron Microscopy Sciences, Hatfield, PA, USA). Specimens were viewed on a Philips CM12 TEM operating at 80 kV.

## Genome library preparation and sequencing

Genomic DNA was extracted from cultivated tachyzoites that were filtered through a 5 μm filter to remove host cells using the QIAamp DNA Mini Kit (QIAGEN) following the manufacturer's protocol. The DNA concentration was measured using a Qubit Fluorometer dsDNA Broad Range assay (Life Technologies). The SMRT bell, was produced using the DNA Template Prep Kit 1.0 (Pacific Biosciences). 5 μg of gDNA were mechanically sheared to an average size distribution of 15–20 kb, using a Covaris gTube (Kbiosciences). A Bioanalyzer 2100 12K DNA Chip assay (Agilent) was used to assess the fragment size distribution. 5 μg of sheared gDNA was DNA damage repaired and end-repaired using polishing enzymes. A blunt end ligation of adapters followed by exonuclease treatment was performed to create the SMRTbell template. A Blue Pippin device (Sage Science) was used to size select the SMRTbell template and enrich the big fragments of > 5 kbp. The size-selected library was quality inspected and quantified using the Agilent Bioanalyzer 12Kb DNA Chip and on a Qubit Fluorimeter (Life technologies) respectively. A ready to sequence SMRT bell-Polymerase Complex was created using the P6 DNA/Polymerase binding kit 2.0 (Pacific Biosciences) according to the manufacturer instructions.

The Pacific Biosciences RS2 instrument was programmed to load and sequence the sample on 14 SMRT cells v3.0 (Pacific Biosciences), taking 1 movie of 240 min each per SMRT cell. A MagBead loading method (Pacific Biosciences p/n 100-133-600) was chosen in order to improve the enrichment of the longer fragments. After the run, a sequencing report was generated for every cell via the SMRT portal in order to assess the adapter dimer contamination, the sample loading efficiency, the obtained average read length and the number of filtered subreads. All sequences are deposited at NCBI's Sequence Read Archive (SRA) under Bioproject PRJNA386239.

## Genome assembly

To remove contamination with human sequences, CCS reads were aligned to the human reference genome (ensembl95) with pbmm2 (version 0.12.0, options–preset CCS -j 6, github.com/PacificBiosciences/pbbioconda) and only reads that did not align were used further. Sequence variants were called in a similar way described previously [42]. Reads were aligned with pbmm2 (options–preset CCS–sort -j 6 and adding a readgroup) and large structural variants were called with pbsv discover and pbsv call (version 2.1.1, default options, github.com/PacificBiosciences/pbbioconda). Homozygous variants were extracted and included in the draft genome with vcf-consensus (vcftools version 0.1.16 [43]). Reads were aligned again to the new draft genome with pbmm2 (options–preset CCS–sort -j 6 and adding a readgroup) and small structural variants, and SNPs were called with GATK (version 4.1.0.0, options–sample-ploidy 2 –pcr-indel-model AGGRESSIVE–read-filter NotSupplementaryAlignmentRead-Filter–minimum-mapping-quality 60 –emit-ref-confidence GVCF–annotation-group AS_StandardAnnotation [44]). Variants were filtered for a minimal genotype quality ("GQ") of 20 with bcftools (version 1.9, github.com/samtools/bcftools). Homozygous variants were extracted and included in the draft genome with vcf-consensus. With the Pacbio data, we included 255 deletions (total length of 9,402 bp), 241 insertions (total length of 8,912 bp), and 104 SNPs. Further variants from the RNA-Seq data were included as described below.

## Gene annotation

First, we identified repeats and potential transposable elements (TEs) with RepeatModeler (version 1.0.11[45]), with NCBI blast 2.6.0+[46]) and RepeatMasker (version 4.0.8 [47]). Second, we used RepeatMasker (version 4.0.8[47]) again to identify repeats and potential transposable elements and the *Sarcocystidae* annotation available from RepBase (RepBaseRepeatMaskerEdition 20181026, www.girinst.org).

To transfer the annotation from the Ger1 reference assembly to the Lis-4 genome, we used flo [48]. In total, 8,224 out of 8,550 were transferred with the liftover. The remaining 326 genes were non-coding RNAs. To identify potentially novel genes, we used the unmasked reference sequence, the unique alignments from the RNA-Seq samples, and the *T. gondii* proteins (ToxoDB, release 41, strain ME49) in conjunction with BRAKER (version 2.1.2, options—bam = uniqueTachyzoites.bam,uniqueTissueCysts.bam—prot_seq = proteins.fa—prg = gth [49–51]). Genes not overlapping with known genes from the liftover, were added to the final annotation choosing the gene name prefix "BESBLIS14_" (267 additional genes). Proteins were annotated with PANNZER2 [52]. Genes were further annotated with the closest *T. gondii*, *E. tenella*, *N. caninum*, and *P. falciparum* homologues. We therefore aligned the protein sequences to a collection of *T. gondii*, *E. tenella*, *N. caninum*, and *P. falciparum* proteins with DIAMOND reporting up to ten alignments with a maximal e-value of 0.00001 (version 0.9.14, options -e 0.00001 -k 10 [53]). The protein with the highest alignment score was then assigned as homologue.

## Synteny, phylogeny and orthology analysis

Synteny analysis was performed with MCScanX using default values (most recent binary retrieved in May 2019 [54]) by the pairwise determination of syntenic regions between species, which were inferred from the order of orthologues. Orthologues between *B. besnoiti* and the two other species *T. gondii* and *N. caninum* were identified with DIAMOND reporting up to five alignments with a maximal e-value of 0.00001 (version 0.9.14, options -e 0.00001 -k 10 [53]). Protein data were taken from the Lis14 annotation (*B. besnoiti*, ToxoDB *T. gondii* ME49, release 52) and the annotation provided for *N. caninum* Liverpool strain by Berná *et al.* [20]. Contigs shorter than 100 kb were removed for plotting (*T. gondii*: 2,251 contigs, 3,185,967 bp; *N. caninum*: 29 contigs, 661,796 bp; *B. besnoiti*: 168 contigs, 1,632,323 bp). Graphs were generated with Circos [55] and edited in Inkscape.

Phylogenetic analysis using whole genome sequences was performed using Co-Phylog [56]. To define orthologous groups, OrthoMCL [57] was performed using version 14–137. We further devised a set of orthologous genes based on the synteny described above. For this, we extracted all syntenic blocks with at least 10 genes and assigned orthology between two species if genes were in the corresponding syntenic blocks. Venn diagrams of orthologous groups were illustrated using the webtool http://bioinformatics.psb.ugent.be/webtools/Venn/.

## *In silico* protein analysis

Alignment of the *T. gondii* and *B. besnoti* HAP2 protein sequences (TGME49_285940 and BESB_068050) using MUltiple Sequence Comparison by Log- Expectation (MUSCLE, https://www.ebi.ac.uk/Tools/msa/muscle/) and Protein BLAST (NCBI - https://blast.ncbi.nlm.nih.gov/). Protein domain predictions were performed using the Simple Modular Architecture Research Tool (SMART, http://smart.embl-heidelberg.de/) in the normal mode selecting searches for outlier homologues, PFAM domains, signal peptides and internal repeats. 3D structure prediction of full-length proteins (pre-proteins) was performed using the I-TASSER online server (https://zhanggroup.org/I-TASSER/).

## RNA extraction, library preparation and sequencing

RNA was extracted from cultivated tachyzoites that were filtered through a 5 μm filter to remove host cells and then cryopreserved in Trizol (Life Technologies) at -80˚C. Tissue cysts isolated from a naturally infected cow was cryopreserved in Trizol (Life Technologies). RNA extraction was performed using the manufacturer's protocol. The RNA was then treated with DNase I (Qiagen DNase I kit), precipitated and washed to remove the DNase. The quality of the isolated RNA was determined with a Qubit (1.0) Fluorometer (Life Technologies) as well as with a Bioanalyzer 2100 (Agilent). Two RNA samples for each stage were pooled for the following steps. The TruSeq RNA Sample Prep Kit v2 (Illumina Inc,) was used in the succeeding steps. Briefly, total RNA samples (100–1000 ng) were poly-A enriched and then reverse-transcribed into double-stranded cDNA. The cDNA samples were fragmented, end-repaired and polyadenylated before ligation of TruSeq adapters containing the index for multiplexing. Fragments containing TruSeq adapters on both ends were selectively enriched with PCR. The quality and quantity of the enriched libraries were validated using Qubit (1.0) Fluorometer and the Caliper GX LabChip GX (Caliper Life Sciences Inc.) and qPCR. The libraries were normalized to 10 nM in 10 mM Tris-Cl pH8.5/0.1% Tween 20. The TruSeq PE Cluster Kit v3-cBot-HS or TruSeq SR Cluster Kit v3-cBot-HS (Illumina Inc.) was used for cluster generation using 10 pM of pooled normalized libraries on the cBOT. Stranded sequencing was performed on a full lane per life cycle stage on the Illumina HiSeq 2000 with paired end sequencing at 2 X125 bp using the TruSeq SBS Kit v3-HS (Illumina Inc.). The raw RNA-Seq reads are deposited in the

Sequence Read Archive with accession number SRS2223086 for bradyzoites and SRS2223087 for tachyzoites; expression levels of genes mapped to the Ger1 genome can also be accessed on ToxoDB.

## Transcriptome assembly

Adaptor sequences and low-quality stretches within the reads were removed with TrimGalore (options–paired–illumina, version version 0.5.0, www.bioinformatics.babraham.ac.uk/projects/trim_galore). Calling variants from RNA-Seq data was done as described in the GATK documentation (https://software.broadinstitute.org/gatk/documentation/article.php?id=3891) as far as possible (i.e., base recalibration required known sites that were not available). Variant calling was done separately for each sample (tachyzoites and tissue cysts). Reads were aligned to the draft genome containing the variants from the PacBio data with STAR (version 2.7.0e, option–twopassMode Basic [58]). Duplicates were marked with Picard tools (version 2.18.25, broadinstitute.github.io/picard). Mapping qualities were reassigned with GATK (version 3.8-1-0, options -rf ReassignOneMappingQuality -RMQF 255 -RMQT 60 -U ALLOW_N_CIGAR_READS[59]). Variants were called with GATK (version 3.8-1-0, options -T HaplotypeCaller -dontUseSoftClippedBases -stand_call_conf 20.0). Variants were filtered with GATK as recommended in the user guide (version 3.8-1-0, options -T VariantFiltration -window 35 -cluster 3 -filterName FS -filter "FS > 30.0" -filterName QD -filter "QD < 2.0"). Variants from the two samples were merged with GATK (version 4.1.0.0). If two variants were conflicting, only the variant with the higher quality ("QD") was kept. Homozygous variants were extracted and included in the draft genome with vcf-consensus. With the RNA-Seq data, we included 714 deletions (average size of 2.32 bp), 2,122 insertions (average size of 1.34 bp), and 2,287 SNPs. For the annotation, trimmed RNA-Seq data were again aligned to the final BbLis14 reference sequence with STAR (version 2.7.0e, option–twopassMode Basic [58]) keeping only the unique alignments.

## Differential expression analysis and gene ontology enrichment

Transcripts were quantified with Salmon (version 0.13.1, options–type quasi -k 31 for the index build step and options–validateMappings -l A for the quantification step [60]) using the cDNA and gene annotation (GTF file) provided by BRAKER. Data were normalized with edgeR (version 3.22.5[61]) and genes with an absolute log2-fold change (LFC) greater than 3 were considered to be differentially expressed.

For comparison between *T. gondii* and *B. besnoiti* transcriptomic data, TPMs from ToxoDB or normalized for *T. gondii* tissue cysts [38] were used. To query number of regulated transcripts in ToxoDB, a fold change of >2 and a floor of 10 reads were set.

## Rapid amplification of cDNA ends (RACE) PCR

Primers used here are listed in S9 Table. RNA extracted as above was subjected to first strand cDNA synthesis using the QuantiTect Reverse Transcriptase kit (QIAGEN) using the manufacturer's instructions, but employing the anchor primer 16 which contains polyT for cDNA synthesis. PCRs were then performed using *B. besnoiti srs44*-specific primer 3677 and primer 12 that contains the identical 5' sequence as primer 16 to sequence the 3' UTR sequence. For analysis of the coding sequence, primers 3734 and 3738 were used. PCRs were performed using the Phusion Hot Start II DNA Polymerase (ThermoFisher Scientific). Bands were excised from a 0.8%-agarose gel, and DNA was extracted using the Monarch gel extraction kit (New England BioLabs). Purified PCR products were ligated into the pCR Blunt vector using the Zero Blunt PCR Cloning Kit (Invitrogen) and transformed into DH10α or DH10B

*Escherichia coli.* Single clones were sent for Sanger sequencing using the Ecoli NightSeq service by Microsynth. To identify the 3'UTR, Sanger sequencing was performed with primers M13F and M13R. To assess the coding sequence, the inserst were ligated into the pCR Blunt vector and sequenced with M13 F and primers 3744–3749. Sequences alignments were performed using **Mu**ltiple **S**equence **C**omparison by **L**og- **E**xpectation (MUSCLE, https://www.ebi.ac.uk/Tools/msa/muscle/).

## Domain and structure predictions

Alignments were performed using MUSCLE (3.8, https://www.ebi.ac.uk/Tools/msa/muscle. Protein domain prediction were performed using Simple Modular Architecture Research Tool (SMART, http://smart.embl-heidelberg.de/) with searches for outlier homologues, Pfam domains, signal peptides and internal repeats activated. For the prediction of 3D structures, the I-TASSER online server [62–64] was used.

## Comparative in silico analysis of metabolic gene expression

A previous version of the annotated *B. besnoiti* genome (unmasked and masked) was mined for metabolic genes based on orthology to the closely related coccidian *T. gondii*. Tachyzoite and tissue cyst read counts and not dramatically different from those found in the newer annotation, Tab A in S1 Table. All identified genes were classified into 10 metabolic groups and 59 pathways for the production of key metabolites and maintenance of diverse cellular processes (Tab B in S8 Table), based on the recently curated metabolic network for *T. gondii*, iTgo [29]. A total of 692 open-reading-frames (ORFs) could be mapped and assigned to metabolic functions without missing genes in these pathways. All genes were assigned and categorized based on their enzyme classification (EC) numbers. Additionally, all 59 metabolic pathways were sorted based on the expression level of the genes either tachyzoites or bradyzoites. Gene IDs from the newer annotation can be found in Tab B in S8 Table.

## Supporting information

**S1 Fig. Quality control of RNA from *B. besnoiti* tachyzoites and tissue cysts. (A)** Simulated gel images of RNA profiles from Bioanalyzer analyses. Shown are the two samples pooled for RNA-Seq. Red asterisks mark the parasite-specific 26S ribosomal RNA bands, arrows the host-specific 28S rRNA bands and blue arrow heads the parasite- and host-specific 18S rRNA bands. **(B)** Bar graph showing percentage of the unique and non-unique reads mapped to the *B. besnoiti* genome.
(TIF)

**S2 Fig. Structure of *B. besnoiti* BESB_048010. (A)** Gene model and mapped RNA-Seq reads (ToxoDB release 53). **(B)** Predicted transcripts of *B. besnoiti* SRS44, model inferred using RNA-Seq data and a model for 3' end resulting from RACE PCR data and CDS from primer walking. **(C)** Domain structures of *B. besnoiti* and *T. gondii* SRS44.
(TIF)

**S3 Fig. Bar graphs depicting expression value differences (normalized reads [bradyzoite read counts–tachyzoite read counts]) of b. besnoiti ROP, GRA, MIC, and SRS orthologues based on the data in S7 Table.** Genes overexpressed in bradyzoites or in tachyzoites are indicated in orange and blue, respectively.
(TIF)

**S4 Fig. Alignment of the amino acid sequenced of *T. gondii* and *B. besnoiti* HAP2.** Asterisks mark identical, colons very similar, and dots similar amino acids.
(TIF)

**S5 Fig. Structure prediction of *B. besnoiti* HAP2. (A)** Dotplot of the sequence alignment. **(B)** Domain predictions of *B. besnoiti* and *T. gondii* HAP2 using SMART. **(C)** Prediction of 3D structure of *B. besnoiti* and *T. gondii* HAP2 using I-TASSER.
(TIF)

**S1 Table. Gene expression in *B. besnoiti* Lis14.**
(XLSX)

**S2 Table. *B. besnoiti* genes with no evidence for expression.**
(XLSX)

**S3 Table. 5615 orthogroups comprising a total of 9110 genes that contain at least one gene of each of the cyst-forming coccidia.**
(XLSX)

**S4 Table. Significantly higher expressed gene in *B. besnoiti* tachyzoites and tissue cysts.**
(XLSX)

**S5 Table. *B. besnoiti* orthologues of 20 genes coding for factors which have been localized to the tissue cyst wall in *T. gondii*.**
(XLSX)

**S6 Table. *B. besnoiti* synthenic orthologues of *T. gondii* GRA genes.**
(XLSX)

**S7 Table. Members of *B. besnoiti* key secreted protein families, the GRAs, MICs, ROPs, and GPI-anchored SRSs with orthologues in *T. gondii*.**
(XLSX)

**S8 Table. Metabolic pathways and genes identified the genome of *B. besnoiti* and analysis for stage-specificity.**
(XLSX)

**S9 Table. Representation of *B. besnoiti* orthologues within the 612 ORFs of the combined *T. gondii* TgEES-RO gene set.**
(XLSX)

**S10 Table. *B. besnoiti* AP2 orthologues.**
(XLSX)

**S11 Table. Microgamete-specific genes with *B. besnoiti* orthologs.**
(XLSX)

## Acknowledgments

We are grateful do Dr. Andrea Patrignani and Dr. Sirisha Aluri from the Functional Genomics Center Zurich for their support. We thank Luisa Berná, Maria Eugenia Francia and Carlos Robello for a new assembly of the *N. caninum* genome sequence prior to publication. We are thankful for their participation in obtaining the tissue cysts to Reto Rufener, Vera Manser, University of Bern; Helder Cortes, University of Évora; Evaristo Silva, Vet+ Montemor-o-Novo; Afonso Basto, Ana Caldeira, Andreia Ferreira, Dulce Santos, Helena Soares, Inês Delgado, Ricardo Pinto, Sara Zúquete, Sofia Nolasco, University of Lisbon.

## Author Contributions

**Conceptualization:** Chandra Ramakrishnan, Aarti Krishnan, Alexandre Leitão, Andrew Hemphill, Dominique Soldati-Favre, Adrian B. Hehl.

**Data curation:** Chandra Ramakrishnan, Aarti Krishnan, Marc W. Schmid, Giancarlo Russo, Andrew Hemphill, Adrian B. Hehl.

**Formal analysis:** Chandra Ramakrishnan, Aarti Krishnan, Marc W. Schmid, Giancarlo Russo, Alexandre Leitão, Andrew Hemphill, Adrian B. Hehl.

**Funding acquisition:** Dominique Soldati-Favre, Adrian B. Hehl.

**Investigation:** Chandra Ramakrishnan, Aarti Krishnan, Samuel Francisco, Marc W. Schmid, Giancarlo Russo, Alexandre Leitão, Andrew Hemphill, Dominique Soldati-Favre, Adrian B. Hehl.

**Methodology:** Aarti Krishnan, Alexandre Leitão, Andrew Hemphill.

**Project administration:** Chandra Ramakrishnan, Alexandre Leitão, Dominique Soldati-Favre, Adrian B. Hehl.

**Resources:** Samuel Francisco, Alexandre Leitão, Andrew Hemphill, Adrian B. Hehl.

**Software:** Aarti Krishnan, Marc W. Schmid, Giancarlo Russo.

**Supervision:** Chandra Ramakrishnan, Alexandre Leitão, Dominique Soldati-Favre, Adrian B. Hehl.

**Validation:** Chandra Ramakrishnan, Aarti Krishnan, Marc W. Schmid, Giancarlo Russo, Alexandre Leitão, Andrew Hemphill, Adrian B. Hehl.

**Visualization:** Chandra Ramakrishnan, Aarti Krishnan, Samuel Francisco, Marc W. Schmid, Giancarlo Russo, Alexandre Leitão, Andrew Hemphill, Adrian B. Hehl.

**Writing – original draft:** Chandra Ramakrishnan, Aarti Krishnan, Marc W. Schmid, Alexandre Leitão, Andrew Hemphill, Dominique Soldati-Favre, Adrian B. Hehl.

**Writing – review & editing:** Chandra Ramakrishnan, Aarti Krishnan, Alexandre Leitão, Andrew Hemphill, Dominique Soldati-Favre, Adrian B. Hehl.

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
