## [Decision Letter · Decision Letter 0]

29 Jul 2022

Dear Drs A. Hehl and C. Ramakrishnan,

Thank you very much for submitting your manuscript "Dissection of Besnoitia besnoiti intermediate host life cycle stages: from morphology to gene expression" for consideration at PLOS Pathogens. As with all papers reviewed by the journal, your manuscript was reviewed by members of the editorial board and by 3 independent reviewers. 

As part of the lifecycle of Besnoitia besnoiti remains puzzling, the editors and reviewers have appreciated your study representing a useful addition to the field. However, we found that the presentation of the data needs rearrangement and more accuracy in the gene names; the main purpose/message of the work is confusing; and the work will benefit from a better integration of the results with sexual stages of other Apicomplexa.

Sincerely,

Isabelle Coppens

Associate Editor

PLOS Pathogens

Vern Carruthers

Section Editor

PLOS Pathogens

Kasturi Haldar

Editor-in-Chief

PLOS Pathogens

orcid.org/0000-0001-5065-158X

Michael Malim

Editor-in-Chief

PLOS Pathogens

orcid.org/0000-0002-7699-2064

Reviewer Comments (if any, and for reference):

Reviewer's Responses to Questions

**Part I - Summary**

Reviewer #1: A review of genome and RNAseq data on Besnoitia besnoiti isolated from cattle. The analysis of the sequence data is reasonable and is correlated with data in T. gondii and other Apicomplexa. The data indicate that a sexual cycle exists and, to an extent, exclude an insect host (based on looking at Plasmodium homologs that might indicate an insect host). I do not think this is completely excluded by the data, but it is a reasonable suggestion. The light and EM images of the cysts are useful and add to the literature. In fact a more detailed series of images in the supplemental figures could be useful.

Reviewer #2: 1. Here the authors use microscopy, genome sequencing, and transcriptomics to characterize Besnoitia besnoiti in its in vitro tachyzoite and in vivo bradyzoite forms. Genomic comparisons to Toxoplasma gondii, Neospora caninum, Eimeria tenella, and Plasmodium falciparum are made as well as transcriptomic comparisons to T. gondii. In silico evidence for a definitive B. besnoiti host and different metabolic pathways are presented. Overall, a useful addition to the field. Some confusion about the purpose of the paper- is the main idea comparison of intermediate stages as title suggests, or is it more about presenting an argument for a sexual stage and definitive host? To strengthen definitive host argument, deeper analysis of intermediate stage orthologs between T. gondii and B. besnoiti should be performed. Figure confusion needs to be addressed. Recommend some rearrangement and condensation of figures.

Reviewer #3: Besnoitia besnoiti is a cyst-forming apicomplexan protozoan parasite closely related to Toxoplasma gondii that can infect livestock, especially cattle, as intermediate hosts. Bovine besnoitiosis is a disease that progresses in two phases: an acute phase and a chronic phase. The acute phase can be fatal or cause infertility or sterility in males. Symptoms range from fever and anorexia to vascular disorders. In the chronic phase, various skin lesions may occur. Unfortunately, there are no vaccines or therapeutics to date that could halt the progression of the disease. This study represents a milestone in the study of Besnoitia besnoiti biology and provides a wealth of "omics" data on two different stages of the intermediate host cattle. Transmission electron microscope (TEM) analysis revealed that organellar structures match well with those of T. gondii, but also remarkable differences in the cyst wall, which has probably adapted to its particular niche in the skin.

A significant part of the life cycle of this parasite remains unknown, and to date no definitive host has been assigned to it. This study provides unequivocal genomic evidence for the presence of genes in Besnoitia besnoiti that are homologous to those of Toxoplasma and known to be specific for the sexual cycle (development of microgametes and macrogametes) in the cat intestine and oocyst formation, supporting the original hypothesis of a definitive carnivorous/scavenger host for this parasite. Regarding the expression of its genome, B. besnoiti has conserved a fingerprint of conserved stage-specific markers, but is also characterized by a specific subtranscriptome. As in Toxoplasma, the differences between the tachyzoite and bradyzoite stages show very distinct signatures.

The entire life cycle of B. besnoiti remains to be elucidated, but this study provides a fairly comprehensive analysis of the biology of zoites responsible for acute and chronic disease.

**Part II – Major Issues: Key Experiments Required for Acceptance**

Reviewer #1: The data is clearly presented and appears complete. I do not see much need for additional validation of the RNAseq data using immunoblot or proteomic approaches; as one would expect a reasonably close correlation between the RNAseq data and results on protein levels for the genes discussed.

The author notes that the CST1 homolog in Besnoitia lacks a mucin domain and would not be a source of lectin staining; however, I did not see any mention of lectin staining of these cysts. It would be useful to provide this data (e.g show that cysts stain with Dolichus). If they are DBA positive then the issue is if this is still due to glycosylation of the CST1 homolog or if another gene exists with a mucin or similar domain that would be DBA positive if post transitionally modified.

Reviewer #2: Regarding in silico evidence for a definitive host: authors rely heavily on existence/number/conservation of B. besnoiti orthologs of Toxo sex genes. To strengthen this argument, why not perform a similar analysis on bradyzoite/cyst genes? Because intermediate lifestage data are available and reliable in both Toxo and B. besnoiti, that analysis should serve as a positive control for what you might expect to observe for sex genes. For example, of the B. besnoiti genes that are highly expressed in Bb cysts relative to tachyzoites, how many are orthologous to Toxo bradyzoite/cyst genes? Add this comparative analysis to figure 6.

Reviewer #3: (No Response)

**Part III – Minor Issues: Editorial and Data Presentation Modifications**

Reviewer #1: I would suggest that Figure S1 on the TEM of the cyst be included in the figures in the paper rather than supplemental data. In addition, if more images exist of the TEM and SEM they could be provided as additional supplemental images.

In Figure 1 higher magnification images of the light microscopy with HE and MB/fuchsin should be provided.

Reviewer #2: Figure legends:

o please specify that images are of the cyst wall surface in Fig. 1D-G

o 7D description is repeated. difference between 7C and D is unclear in legend, figure, and text (more below)

The authors write as though the most interesting part of the paper is evidence for an as-yet unknown definitive host, which is interesting. For example, in the intro and results, evidence for a definitive host is discussed and presented first. Recommend discussing/presenting sexual stage last after discussing intermediate stages.

Page 9: authors note that 432 genes were expressed in neither tachys nor bradys. how does that fraction compare to T. gondii gene expression studies? ie what fraction of T. gondii genes are not expressed in tachys or bradys?

Page 10/figure 4a: recommend adding an intermediate color (or multiple) between blue and red. it is difficult to distinguish the shades of red and blue, blue in particular.

Page 12/figure 6c: there is no figure 6c

Page 12/figure 6: text doesn’t match figure. figure shows highest percentage of genes are EES1, text states highest percentage of genes are EES5 and oocyst

Page 14/figure 7: figures 7c and 7d are confusing. why are there opposite colorations for most genes if the only different between c and d is the scale? I am interpreting “difference to common mean” as only a change in scale, so if that is not correct, the difference between c and d needs to be better explained in text and figure legend.

Page 14/figures 7 and 8: as is, recommend combining and condensing figures 7 and 8. move 7a to supplement and replace with 8a, as they are quite similar. 7b is fine but doesn’t say much: could either go to supplement or have added annotation to describe which classes of genes are high and low in tachys vs. cysts. there is plenty of room to make the colored portion of the plot narrower to make room for labels. 7c/d are very confusing, perhaps remove one? the dot plots in 8c/d are ok, but could easily be communicated as heatmaps like in 7c/d, with one heatmap for Toxo & one for B. besnoiti, side-by-side. 7c/d seem very similar to 8 c/d, just different genes. Given available data for Toxo, recommend adding a Toxo gene expression heatmap next to the 7c B. besnoiti heatmap. So in summary for 7c/d and 8c/d, make all into heatmaps, each with maps of B. besnoiti alongside maps of Toxo.

Reviewer #3: The manuscript contains a large amount of data, some of which is unfortunately inadequately presented. Some rewording is necessary to guide the reader through concepts that may be difficult for those outside the field to understand.

Below are some points to improve the clarity and presentation of the data:

1- Differential expression of SRS genes, ROP, MIC, and GRA genes (including newly discovered/renamed genes) should be represented as in Figure 3 of Hehl et al. (BMC Genomics, 2015). A similar graphical representation could also highlight conservation with Toxoplasma and other CFAs.

2- The authors should better document the conservation of proteins responsible for interaction with host cells, especially the ROP and GRA proteins of B. besnoiti, by specifying their polymorphism, which has often been associated with their function (e.g. ROP16) or stage-specific expression (e.g. ROP18) or even interaction with their partner, e.g. KIM domains of GRA24 as drivers of host MAPK p38-alpha interaction and activation.

3- AP2 factors deserve special attention and should be presented in a scheme with their structural domain (AP2, ACDC) together with their putative Toxoplasma, but also Eimeria orthologs, indicating at which stage they are expressed in these parasites. Identification of MORC-associated AP2 orthologs would provide further evidence that epigenetic mechanisms exist in B. besnoiti to control sexual commitment.

4- To identify genes likely involved in the B. besnoiti sexual cycle, the authors looked for known homologs in Toxoplasma, but what about conservation in Eimeria? Are the 267 transcripts mentioned in the study unique to B. besnoiti or are they conserved in some CFAs?

5- Table S7 should list TgIST, TgNSM, and TEEGR as bona fide proteins that are presents in dense granule.

PLOS authors have the option to publish the peer review history of their article (what does this mean?). If published, this will include your full peer review and any attached files.

Reviewer #1: No

Reviewer #2: **Yes: **Laura Knoll

Reviewer #3: No

Figure Files:

Data Requirements:

Reproducibility:

References:

---

## [Decision Letter · Decision Letter 1]

17 Oct 2022

Dear Dr. Adrian B. Hehl,

Thank you very much for submitting your manuscript "Dissection of Besnoitia besnoiti intermediate host life cycle stages: from morphology to gene expression" for consideration at PLOS Pathogens. As with all papers reviewed by the journal, your manuscript was reviewed by members of the editorial board and by several independent reviewers. The reviewers appreciated the attention to an important topic. Based on the reviews, we are likely to accept this manuscript for publication, providing that you modify the manuscript according to the review recommendations.

For the Editor:

Two rectifications:

1) The morphological identification of organelle: In Fig. 3: It is speculative to identify a vacuolar compartment serving as a Ca2+ storage and sodium transport activity (VAC) in the absence of marker for this organelle. Could sections of progeny. Replace VAC by VAC? The ap for apicoplast in panel C is erroneous. An apicoplast is visible just above the nucleus based on multiple membranes and global shape. The inner membrane complex beneath the plasma membrane can also be shown. The IMC is an organelle.

2) Regarding a cyst wall staining with DAB: the image with Toxoplasma cysts as positive controls must be shown in the manuscript as it has been done. Not as personal communication to the reviewer.

Sincerely,

Isabelle Coppens

Associate Editor

PLOS Pathogens

Vern Carruthers

Section Editor

PLOS Pathogens

Kasturi Haldar

Editor-in-Chief

PLOS Pathogens

orcid.org/0000-0001-5065-158X

Michael Malim

Editor-in-Chief

PLOS Pathogens

orcid.org/0000-0002-7699-2064

For the Editor:

Two rectifications:

1) The morphological identification of organelle: In Fig. 3: It is speculative to identify a vacuolar compartment serving as a Ca2+ storage and sodium transport activity (VAC) in the absence of marker for this organelle. Could sections of progeny. Replace VAC by VAC? The ap for apicoplast in panel C is erroneous. An apicoplast is visible just above the nucleus based on multiple membranes and global shape. The inner membrane complex beneath the plasma membrane can also be shown. The IMC is an organelle.

2) Regarding a cyst wall staining with DAB: the image with Toxoplasma cysts as positive controls must be shown in the manuscript as it has been done. Not as personal communication to the reviewer.

Reviewer Comments (if any, and for reference):

Reviewer's Responses to Questions

**Part I - Summary**

Reviewer #2: The author have adequately addressed the reviewers concerns.

Reviewer #3: This is a timely and well done study. The authors have addressed all the points raised by the reviewer. The work has been greatly improved with the addition of new analysis and tables.

**Part II – Major Issues: Key Experiments Required for Acceptance**

Reviewer #2: (No Response)

Reviewer #3: (No Response)

**Part III – Minor Issues: Editorial and Data Presentation Modifications**

Reviewer #2: (No Response)

Reviewer #3: (No Response)

PLOS authors have the option to publish the peer review history of their article (what does this mean?). If published, this will include your full peer review and any attached files.

Reviewer #2: **Yes: **Laura J. Knoll

Reviewer #3: **Yes: **HAKIMI Mohamed-Ali

Figure Files:

Data Requirements:

Reproducibility:

References:

---

## [Editor Report · Decision Letter 2]

27 Oct 2022

Dear Dr. Adrian Hehl,

We are pleased to inform you that your manuscript 'Dissection of Besnoitia besnoiti intermediate host life cycle stages: from morphology to gene expression' has been provisionally accepted for publication in PLOS Pathogens.

Best regards,

Isabelle Coppens

Associate Editor

PLOS Pathogens

Vern Carruthers

Section Editor

PLOS Pathogens

Kasturi Haldar

Editor-in-Chief

PLOS Pathogens

orcid.org/0000-0001-5065-158X

Michael Malim

Editor-in-Chief

PLOS Pathogens

orcid.org/0000-0002-7699-2064
---

## [Editor Report · Acceptance letter]

9 Nov 2022

Dear Prof. Hehl,

We are delighted to inform you that your manuscript, "Dissection of Besnoitia besnoiti intermediate host life cycle stages: from morphology to gene expression," has been formally accepted for publication in PLOS Pathogens.

Best regards,

Kasturi Haldar

Editor-in-Chief

PLOS Pathogens

orcid.org/0000-0001-5065-158X

Michael Malim

Editor-in-Chief

PLOS Pathogens

orcid.org/0000-0002-7699-2064